# Towards Multimodal Data-Driven Scientific Discovery Powered by LLM Agents

**Fan Liu[1], Xiaozhao Zeng[1], Hao Liu[1]***

[1]Hong Kong University of Science and Technology (GuangZhou)
`{fliu236,xzeng638}@hkust-gz.edu.cn`
`liuh@ust.hk`

## Abstract

Recent advances in large language models (LLMs) have enabled agents that automate scientific discovery by interpreting data, generating analysis pipelines, and executing them with computational tools. However, existing benchmarks remain largely limited to unimodal datasets and slice-level tasks, overlooking the fact that real discovery requires multimodal integration, modeling, and hypothesis-driven reasoning. To address this gap, we introduce `MoSciBench`, the first benchmark for multimodal scientific discovery constructed from peer-reviewed studies through a principled four-stage pipeline. `MoSciBench` spans six scientific domains, seven data modalities, and five categories of discovery questions, yielding 88 individual, end-to-end, data-driven tasks. Each task is designed as a cross-modal hypothesis verification workflow, requiring agents to align and integrate heterogeneous datasets before modeling and reasoning. We further evaluate four representative agent frameworks across multiple LLM families. Results show that multimodal discovery is substantially harder than unimodal tasks: even the strongest agents achieve only 48.94% accuracy, with over 60% of failures due to cross-modal alignment. Lightweight workflow scaffolding consistently improves performance, reducing alignment errors by 5–10% and raising accuracy by 5.7% on average. Our benchmark and evaluation framework thus establish a rigorous testbed for advancing LLM agents toward realistic, multimodal scientific discovery. Our code and data are available at https://github.com/usail-hkust/MoSciBench

## 1 Introduction

Scientific discovery is increasingly data-driven, requiring integration of multimodal data (e.g., satellite imagery, climate time series, and tabular measurements), building models to uncover patterns (e.g., predicting extreme climate events or identifying molecular interactions), and validating hypotheses through iterative analysis Li et al. (2025). Traditionally, constructing such end-to-end workflows, from data preparation to model validation, has been manual and expertise-intensive, limiting scalability Zheng et al. (2025a;b). Recent advances in LLMs suggest a new paradigm: agents that can interpret diverse data types, automatically generate analysis pipelines, and execute them with scientific tools Guo et al. (2024); Liu et al. (2025c); Lai et al. (2025); Yuan et al. (2026). Realizing this vision, however, systematic evaluation is needed on realistic multimodal scientific tasks.

Existing benchmarks (e.g., ScienceAgentBench Chen et al. (2024b), DiscoveryBench Majumder et al. (2024)) have advanced LLM-based discovery by formalizing workflows Zhang et al. (2025); Tang et al. (2023); Lu et al. (2024) (e.g., dataset preparation, analysis, model design, and validation). In these benchmarks, however, each task is tied to a single type of dataset, for instance, tabular records or a single time series, so agents are only evaluated within isolated modalities Gu et al. (2024). As a result, they remain restricted to unimodal data (e.g., image, time series, or tabular formats). In addition, many tasks are defined at the level of individual points or slices, where agents handle small fragments, lacking the realism of repository-level discovery Tian et al. (2024). By contrast, real scientific discovery is inherently multimodal, requiring agents to access full repositories, integrate heterogeneous files, and reason across them to generate insights, as illustrated in Figure 1. For

---

*Correspondence to Hao Liu.

instance, climate studies combine satellite imagery with spatiotemporal metadata Liu & Yao (2024), and health research links physiological signals with environmental measures Anders et al. (2024). Capturing this complexity in benchmarks is challenging Zheng (2025), as it requires evaluating agents on cross-modal alignment, modeling, and reasoning capabilities that are essential for practical scientific discovery but largely absent from current benchmarks.

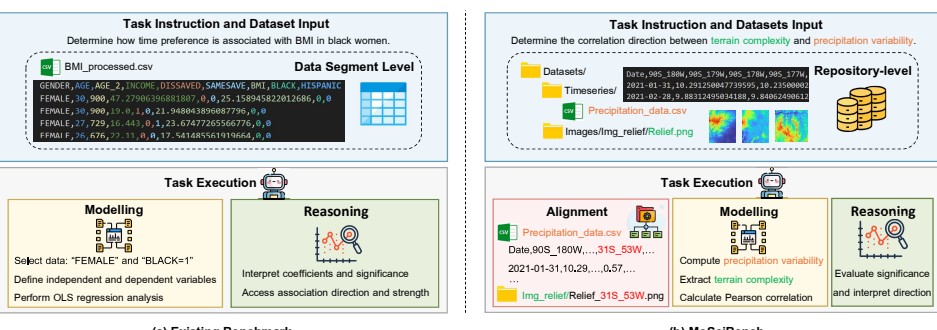

Figure 1: Previous benchmarks vs. `MoSciBench` (ours). **Left:** Existing benchmarks focus on unimodal, small-scale tasks (e.g., single tables or short sequences), offering only fragmented evaluations. **Right:** `MoSciBench` supports end-to-end multimodal discovery by letting agents access full repositories, integrate heterogeneous data, generate and run code, and reason over results to verify scientific hypothesis.

In this work, we introduce `MoSciBench`, the first benchmark for multimodal data-driven scientific discovery. To ensure realism, we construct tasks from peer-reviewed studies through a principled four-stage pipeline: (1) raw data extraction from published repositories, (2) multimodal processing and alignment to clean, standardize, and integrate heterogeneous datasets, (3) task instruction formulation and annotation to encode discovery objectives in executable and verifiable forms, and (4) human verification and quality control to guarantee consistency and reliability. Each task is designed around a scientific discovery goal that requires agents to perform cross-modal alignment, modeling, and reasoning, rather than isolated slice-level predictions. For example, a climate task integrates satellite imagery with numerical storm tracks to assess cyclone intensity, while a health task combines physiological and environmental signals to identify risk factors for cardiac stress. Overall, `MoSciBench` covers six scientific domains (climate science, biomedical engineering, cheminformatics, health psychology, population genomics, and earth science), five categories of discovery questions (descriptive analysis, correlation, causal inference, prediction, and pattern discovery), and seven data modalities (multi-sensor time series, tabular data, satellite imagery, mass spectra, molecular structures, genotype matrices, and HDF matrices), yielding 88 individual tasks in total.

To systematically evaluate LLM agents on scientific discovery, we present `MoSciBench` and a comprehensive evaluation framework. Our contributions are threefold:

❶ **Benchmark.** We introduce `MoSciBench`, the first benchmark for *multimodal scientific discovery*, spanning six domains, seven modalities, and 88 individual tasks. Built from peer-reviewed studies through a principled pipeline (data acquisition, multimodal processing, task annotation, and multi-pass verification), `MoSciBench` explicitly targets multimodal, repository-level discovery, making tasks substantially more complex and realistic than prior unimodal benchmarks.

❷ **Task formalization.** Each task is defined as a *cross-modal hypothesis verification workflow*, where agents must load, preprocess, align, and integrate heterogeneous datasets before modeling and reasoning. Tasks cover five categories central to discovery, descriptive analysis, correlation testing, causal inference, predictive modeling, and pattern discovery, explicitly enforcing multimodal alignment (e.g., linking imagery with time series, or genotype matrices with phenotypes).

❸ **Evaluation.** We systematically evaluate four representative agent frameworks, combined with both open- and closed-source LLM families, on all 88 tasks. Results reveal three findings: (i) **multimodal discovery is significantly harder than unimodal tasks**, with even the strongest agents achieving only 48.4 % accuracy; (ii) **cross-modal alignment is the dominant bottleneck**, accounting for over 60% of errors; and (iii) **lightweight workflow scaffolding consistently boosts performance**, reducing alignment errors by 5–10% and raising accuracy by 5.7% on average.

## 2 MOSCIBENCH CONSTRUCTION

In this section, we introduce `MoSciBench`, a benchmark designed to evaluate LLM agents on multimodal data-driven scientific discovery tasks. These tasks require integrating information from multiple modalities through multimodal data exploration, scientific computation, and reasoning with LLM agents, ultimately aiming to validate scientific hypotheses.

### 2.1 PROBLEM FORMULATION

`MoSciBench` evaluates LLM agents on multimodal data-driven discovery tasks, each framed as an end-to-end workflow requiring cross-modal alignment, scientific modeling, and hypothesis verification. Each task is instantiated with three components: (i) a *task instruction* derived from a peer-reviewed study, specifying the scientific background and hypothesis to be tested; (ii) one or more *multimodal datasets* (e.g., imagery, time series, tabular records, molecular structures) providing the evidence base; and (iii) an *evaluation protocol* that checks whether the agent's output is consistent with the gold-standard hypothesis. To solve a task, the agent must autonomously align and fuse heterogeneous data sources, build models, perform computations, and reason over the results to test the hypothesis.

**Task Instructions.** Each task is framed as a scientific question with three elements: the *background* (e.g., the role of precipitation analysis in climatology), the *hypothesis* to be verified (e.g., identifying the region with the highest average precipitation during 2021–2023), and the expected *answer format*, as shown in Table 1. The answer format provides the key anchor for evaluation, specifying how the hypothesis should be expressed and verified, for instance, categorical outputs (e.g., true/false, class labels), numerical values (e.g., averages, coefficients), short strings, or structural patterns. Instructions are concise and open-ended to encourage agents to autonomously decide on exploration, preprocessing, analysis, or modeling steps. Some tasks include optional *domain knowledge* (e.g., definitions, formulas, methodological hints) to reduce ambiguity without prescribing solutions.

**Multimodal Datasets.** Tasks are grounded in datasets spanning seven modalities: (1) time series from multi-sensor streams (e.g., physiological or climate records), (2) tabular data (e.g., survey results or experimental measurements), (3) satellite imagery (e.g., remote sensing products), (4) mass spectra (e.g., metabolomics or proteomics assays), (5) molecular structures (e.g., chemical compounds), (6) genotype matrices (e.g., population genomics data), and (7) HDF matrices (e.g., high-dimensional simulation outputs). Each dataset is released in a structured directory with previews to expose available variables and formats. Because the dataset is multimodal, agents must not only load and preprocess data but also align heterogeneous modalities, for example, linking satellite imagery with spatiotemporal metadata or integrating physiological signals with environmental variables, before conducting scientific analysis.

**Evaluation.** `MoSciBench` adopts a hypothesis-centered evaluation: an agent is judged by whether its output correctly verifies the hypothesis. The ground-truth hypotheses and answers are derived from peer-reviewed publications, ensuring objectivity and scientific validity. Each task specifies an expected *answer format*, which defines how correctness is assessed. Performance is measured by *exact match accuracy*: a prediction is correct only if it exactly matches the reference answer. For categorical outputs (e.g., strings, integers, class labels), exact identity is required. For numerical values, lists, or coordinates, task-specific tolerances are applied (e.g., numeric precision or ordering) to ensure fairness. This setup makes evaluation automatic, objective, and reproducible. Results are aggregated across tasks and reported as overall accuracy.

### 2.2 DATA COLLECTION AND TASK ANNOTATION

`MoSciBench` is constructed directly from peer-reviewed scientific publications spanning six domains (e.g., climate science, biomedical engineering, and cheminformatics). Each benchmark instance follows a principled four-stage pipeline: (i) raw data extraction, (ii) multimodal processing and alignment, (iii) task instruction formulation and annotation, and (iv) human verification and quality control. The overall pipeline construction of `MoSciBench` is illustrated in Figure 2.

**Raw Data Extraction.** Papers releasing datasets under permissive licenses and posing explicit data-driven scientific questions suitable for hypothesis verification are first selected. From these sources, multimodal datasets, such as imagery, time series, and tabular measurements, are extracted, along

**Table 1:** Representative examples of the five task categories in MoSciBench. Each example highlights a distinct reasoning type required for multimodal scientific discovery.

| Category | Representative Task Instruction | Success Criteria (Example Output from Benchmark) |
|---|---|---|
| Descriptive Analysis | Compute the average precipitation in 2021–2023 across all stations and identify the wettest region. | Output: `[(3, -78)]` |
| Correlational Study | Test the time series correlation between air temperature and shortwave radiation (2021–2023). | Output: `p-value = 0.174` (not significant) |
| Causal Inference | Assess whether a reduction in shortwave radiation leads to decreased precipitation. | Output: `answer: {false}` |
| Predictive Modeling | Predict future daily temperature using the past 30 days of data. | Output: `Best RMSE = 3.8208 (Ridge Regression)` |
| Pattern Discovery | Determine the global trend in heatwave-affected areas during 2021–2023. | Output: `Trend = upward` |

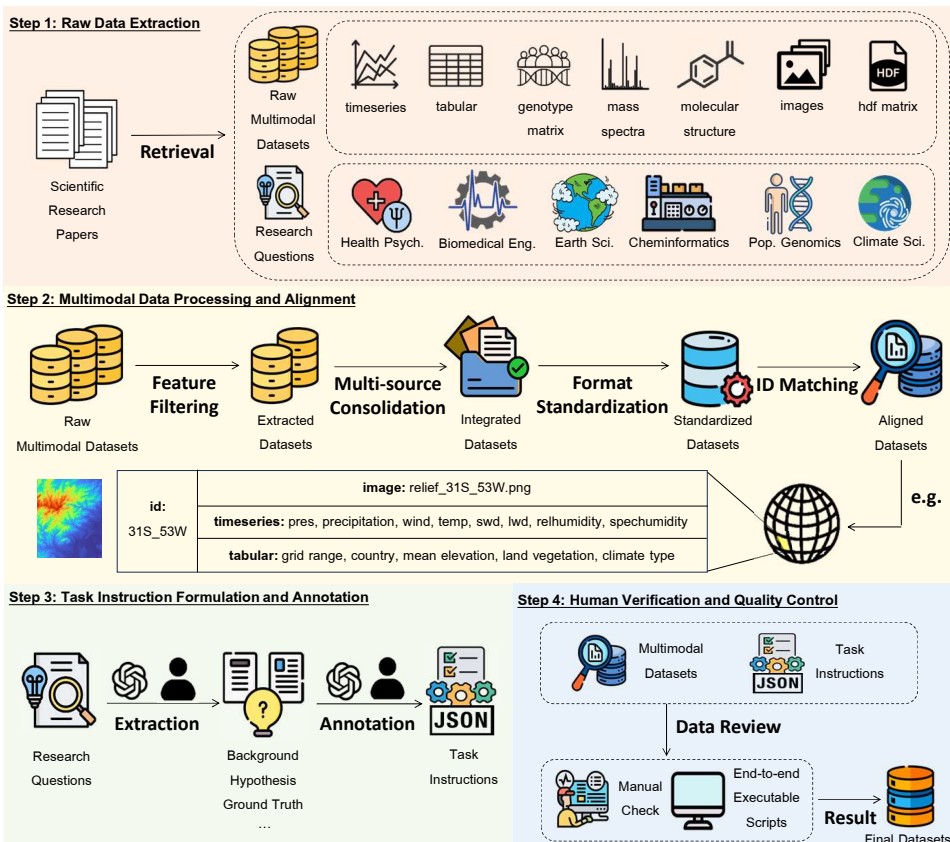

**Figure 2:** Overview of pipeline construction in `MoSciBench`.

with dataset provenance, licensing information, and concise summaries of the associated scientific questions. Dataset descriptors, including variable names and spatial and temporal coverage, are registered in a metadata sheet to support downstream processing. The details of the data collection source and license can be seen in Appendix A.1.

**Multimodal Data Processing and Alignment.** The processing pipeline begins with feature filtering to remove missing or anomalous values and extract a consistent subset of samples from raw multimodal datasets. Next, multi-source consolidation integrates heterogeneous inputs, which are then standardized to ensure comparability by harmonizing units, timestamps, and spatial references. Finally, multimodal alignment is achieved through shared indices: for example, linking individual attributes with physiological time series via subject IDs, or aligning satellite imagery with environmental variables using geographic grids. This step yields aligned datasets, enabling the verification of scientific hypotheses through data-driven downstream tasks.

**Task Instruction Formulation and Annotation.** Each research question is translated into a concise task instruction that preserves the original scientific intent while avoiding overly prescriptive steps. Tasks are paired with verifiable hypotheses, gold answer specifications, and explicit answer formats (e.g., slot-filling, true/false, categorical label). Minimal domain knowledge snippets, such as definitions, formulas, or methodological pointers, are added where necessary to clarify terminology without revealing solutions. Further details of the instructions' composition are provided in Appendix A.1.

**Human Verification and Quality Control.** To ensure task quality, each task instance in `MoSciBench` undergoes multi-pass verification. Verification relies solely on released datasets, ensuring hypotheses can be tested without external resources and multimodal alignments remain correct. In addition to manual checks, annotators implement end-to-end executable scripts that reproduce workflows and automatically check consistency with gold hypotheses, for example validating numerical results within tolerance, checking correlations or causal directions, and assessing predictive performance. Tasks in which human verification conflicted with the original gold-standard hypotheses were filtered out, ensuring that the final benchmark maintains perfect consistency between verified annotations and ground-truth hypotheses (100% agreement).

## 2.3 STATISTICS INFORMATION

**Overall Coverage.** `MoSciBench` is designed around five fundamental categories of data-driven scientific discovery questions, with a total of 88 instantiated subtasks. To highlight the breadth of coverage, we summarize domains, modalities, and task counts in Table 2. These subtasks span six major scientific domains, climate science, biomedical engineering, cheminformatics, health psychology, population genomics, and earth science and incorporate seven complementary data modalities, including multi-sensor time series, tabular data, satellite imagery, molecular structures, mass spectra, genotype matrices, and HDF matrices.

**Table 2: Summary of domains, modalities, and task numbers in `MoSciBench`.** Each domain contains multiple modalities, reflecting the diversity of scientific problems in the benchmark.

| Domain | Modalities | Task Num. |
|---|---|---|
| Climate Science | HDF (matrix), Tabular, Timeseries | 14 |
| Biomedical Eng. | Timeseries, Text | 17 |
| Cheminformatics | Mass spectra, Mol. structures, Tabular | 15 |
| Health Psych. | Timeseries, Tabular | 15 |
| Pop. Genomics | Genotype matrix, Tabular | 13 |
| Earth Sci. | Image, Tabular, Timeseries | 14 |

**Task Categories.** The five categories in `MoSciBench` capture the major reasoning needs of data-driven science: (1) *descriptive analysis* (e.g., summary statistics), (2) *correlational studies* (e.g., correlation tests), (3) *causal inference* (e.g., causal relationship analysis), (4) *predictive modeling* (e.g., regression or classification), and (5) *pattern discovery* (e.g., clustering or factor analysis), as illustrated in Figure 3. Each of the 88 tasks is grounded in one of these categories, providing a relatively balanced distribution across reasoning types. The six domains are evenly integrated, ensuring broad coverage of scientific modalities such as time series, tabular data, molecular structures, and remote sensing imagery. The scale of 88 tasks is deliberately chosen to balance breadth and feasibility: each task is framed as an *end-to-end, repository-level workflow*, where agents must independently perform data-driven computation and reasoning. Given that even a single predictive modeling task can require hours to complete, this design ensures the benchmark remains both challenging and practically executable.

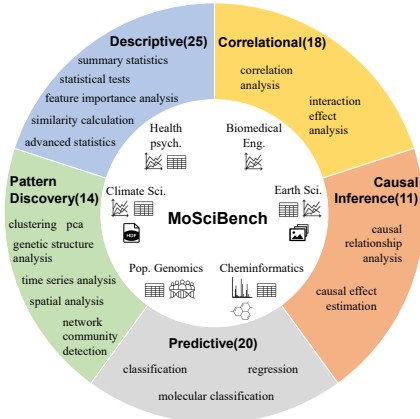

**Figure 3: Distribution of task categories and domains in `MoSciBench`.**

## 3 EXPERIMENTS

In this section, we conduct experiments to address the following research questions: (**RQ1**): How well do current LLM agents perform end-to-end multimodal data-driven discovery on `MoSciBench`? (**RQ2**): What are the main error sources in multimodal scientific discovery, and how can LLM agents be enhanced to address them? (**RQ 3**): What factors influence the performance–efficiency trade-offs of LLM agents in multimodal scientific discovery?

## 3.1 DISCOVERY AGENT

**LLMs and Setup.** We evaluate both open and closed-source models, including `Qwen3-30B-A3B`, `DeepSeek-V3.1`, `gpt-5-mini`, and `o4-mini`. All models are run under a unified configuration with temperature set to 0.0 and zero-shot prompting via API. To prevent excessive computation, the maximum code execution time of each individual task is limited to 1 hour, after which the process is automatically terminated. We report the results of additional model experiments in Appendix A.2.2.

**Agent Frameworks.** Since there are currently no multimodal discovery agents, we follow widely used single-domain discovery agents Majumder et al. (2024); Chen et al. (2024b) and adapt them to the multimodal scientific discovery setting: (1) NoDataGuess: A naive baseline without data-driven methods. It only provides task descriptions and relies entirely on the LLM's internal memory and reasoning ability. (2) ReAct Yao et al. (2023): Alternates between reasoning steps and code execution in an iterative loop to refine hypotheses. (3) DataVoyager: Employs a modular pipeline with planner, code generator, analysis, and critic components to orchestrate discovery. (4) Reflexion (Oracle): Extends CodeGen with oracle feedback and iterative retries (up to three times) for self-improvement. (5) SelfDebug: A program-execution agent that iteratively debugs itself by inspecting execution traces, detecting failure symptoms, and rewriting code based on internal error hypotheses. (6) RAG-ReAct: A retrieval-augmented variant of ReAct that supplements the agent's reasoning trajectory with external domain knowledge. Extended baseline descriptions and a more detailed analysis of experimental patterns across scientific domains are provided in Appendix A.2.4.

**Metric.** We evaluate agent performance using three complementary metrics. (1) **Accuracy.** (*Acc*) A prediction is considered correct only when it exactly matches the reference answer. This strict criterion applies to all categorical outputs, such as numbers, strings, lists, and coordinates. (2) **Code Execution Success Rate** (*Exec*). Measures whether the agent-generated code executes without errors. (3) **Modeling Rationality** (*MR*). A 1–5 LLM-as-judge score assessing the scientific soundness of the workflow, including variable selection, model design, and analytical reasoning. In practice, we use `gpt-4o-mini` as the judge model.

## 3.2 MAIN EXPERIMENTS (RQ1)

Table 3 summarizes the performance of four LLM agents across six domains and 88 tasks. Our observations (Obs.) are as follows: **Obs.❶ LLM agents perform poorly on multimodal data-driven discovery.** Overall accuracy remains modest across all settings, rarely exceeding 50%. Even the strongest configuration, `o4-mini` with ReAct (48.4%) and Reflexion (45.8%), fails to achieve reliable performance. Accuracy is particularly low for smaller models such as `Qwen3-30B-A3B` (best 23.3%) and `gpt-5-mini` (best 17.4%), underscoring the fundamental difficulty of multimodal reasoning and the limitations of current agents in handling multimodal data at scale. **Obs.❷ Data-driven approaches are indispensable.** The non–data-driven baseline (NODATAGUESS) consistently collapses to near-zero accuracy: 0.0% for `Qwen3-30B-A3B` and 2.6% for `DeepSeek-V3.1`, and only 10.5% for `o4-mini`. By contrast, data-grounded frameworks achieve substantially higher scores, with improvements of 20-40 % across six domains. For example, `DeepSeek-V3.1` with ReAct reaches 36.5% and `o4-mini` with Reflexion 45.8%. These results confirm that pure reasoning without access to underlying data is ineffective, while explicit data grounding is critical for meaningful discovery. **Obs.❸ Stronger base models yield stronger agents.** Performance scales directly with the underlying LLM's capability. The strongest model, `o4-mini`, achieves the best overall averages (48.9% with ReAct, 46.6% with Reflexion), while `DeepSeek-V3.1` delivers mid-tier performance (36.5% with ReAct), and `gpt-5-mini` lags far behind (17.4%). This consistent trend indicates that advances in LLM translate directly into more capable downstream scientific discovery agents, reinforcing the tight coupling between base model strength and effective multimodal reasoning.

## 3.3 ERROR ANALYSIS AND AGENT ENHANCEMENT (RQ 2)

**Error Analysis.** To characterize agent failures in multimodal scientific discovery, we classify errors into three categories: *alignment* (conceptual or implementation misalignments), *modeling* (representation, planning, or computation errors), and *reasoning* (statistical or logical inference errors). These categories collectively span the entire LLM agent workflow. We conduct a detailed analysis to uncover underlying causes and common failure modes, as summarized in Figure 4. Our analysis focuses on the best-performing agent, ReAct with the base model

**Table 3: Performance comparison of LLM-based agents across domains.** We report Accuracy / Code Execution Success / Modeling Rationality (MR) for six scientific domains. "Overall" refers to the macro-average across domains.

| Method | Climate Sci. | Biomedical Eng. | Cheminformatics | Health Psych. | Pop. Genomics | Earth Sci. | Overall |
|---|---|---|---|---|---|---|---|
| | (Acc / Exec / MR) | (Acc / Exec / MR) | (Acc / Exec / MR) | (Acc / Exec / MR) | (Acc / Exec / MR) | (Acc / Exec / MR) | (Acc / Exec / MR) |
| | | | | Qwen3-30B-A3B | | | |
| NoDataGuess | 0.000 / – / – | 0.000 / – / – | 0.000 / – / – | 0.000 / – / – | 0.000 / – / – | 0.000 / – / – | **0.000 / – / –** |
| ReAct | 0.143 / 1.000 / 3.64 | 0.412 / 1.000 / 3.41 | 0.333 / 1.000 / 3.67 | 0.133 / 1.000 / 3.50 | 0.308 / 1.000 / 3.67 | 0.071 / 1.000 / 3.77 | **0.233 / 1.000 / 3.58** |
| DataVoyager | 0.000 / 0.364 / 3.58 | 0.529 / 0.600 / 3.62 | 0.067 / 0.875 / 3.79 | 0.067 / 0.909 / 3.47 | 0.154 / 0.091 / 3.85 | 0.071 / 0.778 / 3.55 | **0.148 / 0.603 / 3.64** |
| Reflexion | 0.143 / 0.800 / 3.82 | 0.000 / 1.000 / 4.00 | 0.133 / 1.000 / 3.40 | 0.133 / 0.833 / 3.64 | 0.154 / 0.545 / 4.00 | 0.071 / 0.500 / 3.82 | **0.106 / 0.780 / 3.78** |
| SelfDebug | 0.429 / 0.286 / 3.50 | 0.588 / 0.857 / 3.59 | 0.267 / 0.800 / 3.67 | 0.267 / 1.000 / 3.67 | 0.385 / 0.125 / 3.85 | 0.286 / 0.500 / 3.57 | **0.348 / 0.595 / 3.60** |
| RAG-ReAct | 0.500 / 1.000 / 3.86 | 0.471 / 1.000 / 3.53 | 0.400 / 1.000 / 3.47 | 0.267 / 1.000 / 3.27 | 0.385 / 1.000 / 4.00 | 0.214 / 1.000 / 3.71 | **0.373 / 1.000 / 3.64** |
| | | | | DeepSeek-V3.1 | | | |
| NoDataGuess | 0.000 / – / – | 0.000 / – / – | 0.000 / – / – | 0.000 / – / – | 0.000 / – / – | 0.000 / – / – | **0.000 / – / –** |
| ReAct | 0.429 / 1.000 / 3.64 | 0.647 / 1.000 / 3.71 | 0.400 / 1.000 / 3.73 | 0.267 / 1.000 / 3.33 | 0.308 / 1.000 / 3.69 | 0.143 / 1.000 / 3.64 | **0.365 / 1.000 / 3.63** |
| DataVoyager | 0.286 / 0.600 / 3.71 | 0.529 / 1.000 / 3.82 | 0.333 / 0.750 / 3.60 | 0.267 / 1.000 / 3.67 | 0.154 / 0.545 / 3.69 | 0.143 / 0.625 / 3.50 | **0.285 / 0.753 / 3.67** |
| Reflexion | 0.357 / 0.833 / 3.67 | 0.471 / 1.000 / 3.53 | 0.267 / 1.000 / 3.62 | 0.133 / 1.000 / 3.40 | 0.231 / 0.833 / 3.64 | 0.143 / 1.000 / 3.57 | **0.267 / 0.944 / 3.59** |
| SelfDebug | 0.643 / 1.000 / 3.64 | 0.529 / 1.000 / 3.71 | 0.200 / 1.000 / 3.73 | 0.333 / 1.000 / 3.67 | 0.308 / 1.000 / 3.62 | 0.357 / 1.000 / 3.57 | **0.395 / 1.000 / 3.66** |
| RAG-ReAct | 0.500 / 1.000 / 3.86 | 0.529 / 1.000 / 3.71 | 0.100 / 1.000 / 3.87 | 0.333 / 1.000 / 3.67 | 0.308 / 1.000 / 3.77 | 0.286 / 1.000 / 3.79 | **0.343 / 1.000 / 3.78** |
| | | | | gpt-5-mini | | | |
| NoDataGuess | 0.000 / – / – | 0.088 / – / – | 0.033 / – / – | 0.000 / – / – | 0.000 / – / – | 0.036 / – / – | **0.026 / – / –** |
| ReAct | 0.071 / 1.000 / 3.86 | 0.324 / 1.000 / 3.94 | 0.167 / 1.000 / 3.93 | 0.033 / 1.000 / 3.93 | 0.269 / 1.000 / 3.92 | 0.179 / 1.000 / 3.93 | **0.174 / 1.000 / 3.92** |
| DataVoyager | 0.000 / 1.000 / 4.00 | 0.176 / 1.000 / 4.00 | 0.000 / 1.000 / 4.00 | 0.000 / 1.000 / 4.00 | 0.154 / 1.000 / 4.00 | 0.179 / 1.000 / 4.00 | **0.085 / 1.000 / 4.00** |
| Reflexion | 0.000 / 1.000 / 3.86 | 0.118 / 1.000 / 3.89 | 0.067 / 1.000 / 4.00 | 0.000 / 1.000 / 4.00 | 0.077 / 1.000 / 4.00 | 0.107 / 1.000 / – | **0.061 / 1.000 / 3.95** |
| SelfDebug | 0.143 / 1.000 / 3.92 | 0.353 / 1.000 / 3.94 | 0.000 / 1.000 / 4.00 | 0.000 / 1.000 / 4.00 | 0.154 / 1.000 / 3.85 | 0.214 / 1.000 / 4.00 | **0.144 / 1.000 / 3.95** |
| RAG-ReAct | 0.357 / 1.000 / 3.93 | 0.118 / 1.000 / 4.00 | 0.067 / 1.000 / 4.00 | 0.000 / 1.000 / 4.00 | 0.154 / 1.000 / 4.00 | 0.071 / 1.000 / 4.00 | **0.128 / 1.000 / 3.99** |
| | | | | o4-mini | | | |
| NoDataGuess | 0.143 / – / – | 0.000 / – / – | 0.200 / – / – | 0.067 / – / – | 0.077 / – / – | 0.143 / – / – | **0.105 / – / –** |
| ReAct | 0.571 / 1.000 / 3.50 | 0.647 / 1.000 / 3.76 | 0.333 / 1.000 / 3.93 | 0.533 / 1.000 / 3.80 | 0.462 / 1.000 / 3.92 | 0.357 / 1.000 / 3.57 | **0.484 / 1.000 / 3.75** |
| DataVoyager | 0.357 / 0.375 / 3.50 | 0.588 / 1.000 / 3.59 | 0.133 / 0.714 / 3.79 | 0.333 / 0.667 / 3.67 | 0.231 / 0.200 / 3.75 | 0.286 / 0.600 / 3.50 | **0.321 / 0.593 / 3.63** |
| Reflexion | 0.429 / 0.690 / 3.50 | 0.706 / 0.880 / 3.50 | 0.400 / 0.609 / 3.77 | 0.467 / 0.565 / 3.80 | 0.462 / 0.289 / 3.67 | 0.286 / 0.500 / 3.67 | **0.458 / 0.589 / 3.65** |
| SelfDebug | 0.429 / 0.864 / 3.57 | 0.647 / 0.833 / 3.71 | 0.333 / 0.600 / 3.87 | 0.400 / 0.571 / 3.93 | 0.385 / 0.125 / 3.77 | 0.429 / 0.667 / 3.79 | **0.437 / 0.610 / 3.77** |
| RAG-ReAct | 0.643 / 1.000 / 3.86 | 0.588 / 1.000 / 3.76 | 0.333 / 1.000 / 3.80 | 0.400 / 1.000 / 3.64 | 0.462 / 1.000 / 3.92 | 0.214 / 1.000 / 3.93 | **0.440 / 1.000 / 3.82** |

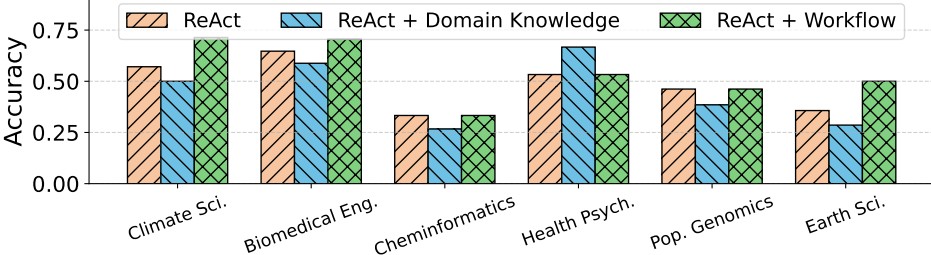

**Figure 5: Performance comparison of ReAct with domain knowledge and workflow scaffolding.** We evaluate three ReAct variants across six domains. Task-provided domain knowledge shows limited gains, whereas lightweight workflow scaffolding enhances, or at least maintains, the capabilities of LLM agents through explicit task decomposition.

`o4-mini`. **Obs.❹ Alignment errors dominate.** The majority of errors are alignment-related (31.8 %), including issues such as information mismatches and data processing failures. The fundamental cause lies in the difficulty of cross-data fusion, linking datasets across domains and transforming diverse forms, distributions, scales, and resolutions into shared, computable representations while preserving domain-specific information. In comparison, modeling errors account for 15.9% and reasoning errors represent only 3.4%. Further details are provided in Appendix A.2.3.

**Agent Enhancement.** We further attempt to enhance LLM agents from two angles: task-provided domain knowledge and lightweight human workflow scaffolding. Specifically, both task-specific domain knowledge and human workflow scaffolding are explicitly incorporated into the agent's context as executable guidance. We evaluate two ReAct variants (ReAct + Domain Knowledge and ReAct + Workflow) across six domains, with results summarized in Figure 5. **Obs.❺ Task-provided domain knowledge yields limited or even negative gains, whereas lightweight workflow scaffolding consistently enhances performance.**

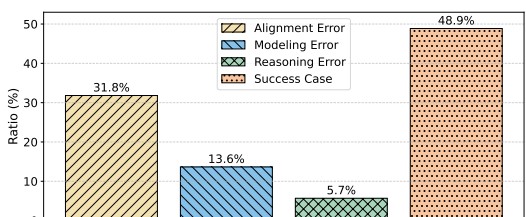

**Figure 4: Error analysis of ReAct.** Distribution of alignment, modeling, and reasoning errors across the multimodal scientific discovery.

On average, ReAct achieves 48.4% across six domains. Incorporating task-provided domain knowledge reduces the average to 44.9%, a decline of 3.5%, with notable drops in climate science (from 57.1% to 50.0%) and cheminformatics (from 33.3% to 26.7%). This suggests that naïvely injecting domain knowledge may introduce noise or misalignment, thereby hindering effectiveness in automated multimodal discovery. In contrast, lightweight workflow scaffolding increases the average to 54.1%, an overall improvement of 5.7%, with the largest relative gains observed in climate science (from 57.1% to 71.4%) and earth science (from 35.7% to 50.0%). These results are consistent with our error analysis: since most failures stem from alignment issues, explicit task decomposition and validation checkpoints introduced by workflow scaffolding significantly improve alignment ability, thereby stabilizing agent performance in multimodal scientific discovery. Specifically, the proportion of alignment errors drops markedly compared with vanilla ReAct (e.g., from 31.8% to 27.3% ), while the share of successful cases increases correspondingly (e.g., from 53.4% to over 60%). This shift shows that workflow scaffolding reduces misinterpretation and data-handling errors while promoting consistent reasoning, yielding more reliable outcomes across domains.

## 3.4 PERFORMANCE–EFFICIENCY TRADE-OFFS (RQ 3)

**Performance by Problem Type.** We further break down performance across the five categories, as shown in Figure 7. **Obs.➎ Performance varies substantially across problem types.** Agents achieve the highest accuracy on causal inference tasks (81.8%), likely because these tasks are explicitly defined and often reduce to structured hypothesis testing, which LLMs can reliably follow when the causal direction is clear. In contrast, descriptive (52.0%) and predictive tasks (50.0%) show moderate accuracy: while agents handle summarization and straightforward

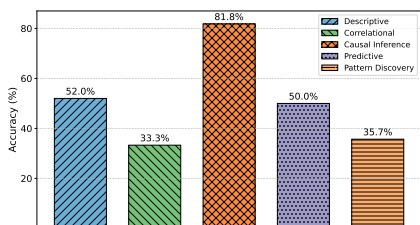

**Figure 7: Performance across problem types.**

supervised modeling, they struggle with maintaining consistency and mitigating error propagation in extended workflows. The sharp drop in correlational (33.3%) and pattern discovery tasks (35.7%) reflects deeper limitations, detecting weak associations or latent structures requires sensitivity to faint statistical signals, robustness to noisy inputs, and inductive generalization beyond observed patterns, areas where current LLM agents remain fragile. Overall, these results suggest that LLMs excel when reasoning steps are well-specified and rule-based, but falter in exploratory tasks that demand subtle statistical rigor, resilience to noise, and open-ended inference.

**Cost Analysis.** We further analyze API costs across both domains and agents. Table 4 summarizes domain-level results, while Table 5 reports agent-level breakdowns. **Obs.➐ Cost-effectiveness varies substantially across domains.** Biomedical Engineering achieves the highest cost-effectiveness (1.1), benefiting from relatively low cost ($0.57) and high task scores (0.65), likely due to the structured nature of biomedical datasets. In contrast, Population Genomics and Earth Science both incur high costs (above $1.0) yet deliver low accuracy (0.46 and 0.36), yielding the lowest CE (0.4). This highlights that domains

**Table 4: Domain-level cost, score, and cost-effectiveness (CE) of Agent ReAct with `o4-mini`.** The table reports results across six scientific domains, highlighting substantial variations in both absolute cost and cost-effectiveness.

| Domain | Cost | Score | CE |
|---|---|---|---|
| **Climate Sci.** | $0.98 | 0.57 | 0.6 |
| **Biomedical Eng.** | $0.57 | 0.65 | 1.1 |
| **Cheminf.** | $0.86 | 0.33 | 0.4 |
| **Health Psych.** | $0.76 | 0.53 | 0.7 |
| **Pop. Gen.** | $1.15 | 0.46 | 0.4 |
| **Earth Sci.** | $1.02 | 0.36 | 0.4 |

with large, noisy, or high-dimensional modalities (e.g., genotype matrices or geoscientific data) are less efficiently handled by current agents. **Obs.➑ Agents exhibit distinct cost–performance trade-offs.** As shown in Table 5, NoDataGuess achieves the lowest cost ($0.04) but offers negligible utility. Reflexion is the most expensive agent ($1.34 on average), driven by repeated trial-and-error loops, yet its performance gains are often marginal relative to the extra cost. ReAct ($0.89) and DataVoyager ($0.77) lie between these extremes: ReAct generally provides higher accuracy but at higher cost, while DataVoyager achieves more balanced efficiency, avoiding Reflexion's overhead while still improving over naive baselines. These results suggest that improving workflow-level efficiency may yield greater gains than simply scaling model size or computation budgets.

**Table 5: Cost comparison across agents with base model `o4-mini`.** The table reports agent-level costs for four representative agents (NoDataGuess, ReAct, DataVoyager, and Reflexion) over six scientific domains. Results highlight substantial cost differences between lightweight baselines and advanced agent strategies under the same base model configuration.

| Agent | Clim. | Bio. | Chem. | Psych. | Gen. | Earth | Avg |
|---|---|---|---|---|---|---|---|
| NoDataGuess | $0.04 | $0.05 | $0.04 | $0.05 | $0.04 | $0.04 | **$0.04** |
| ReAct | $0.98 | $0.57 | $0.86 | $0.76 | $1.15 | $1.02 | **$0.89** |
| DataVoyager | $0.94 | $0.54 | $0.50 | $0.78 | $0.90 | $0.92 | **$0.77** |
| Reflexion | $1.63 | $1.06 | $0.83 | $1.21 | $2.41 | $0.92 | **$1.34** |

**Inference Time Computation.** We investigate whether increasing inference-time computation improves agent performance through two strategies. First, *Best-of-$N$* with ReAct (`DeepSeek-V3.1`) shows gains up to $N = 3$ but declines thereafter, as additional generations increasingly amplify erroneous outputs. Second, Reflexion (Oracle) with `o4-mini` also improves with limited retries but quickly plateaus. **Obs.❾ Inference-time scaling yields diminishing returns.** Because data-driven scientific discovery tasks are inherently complex, each rollout carries a high risk of errors. With a small number of rollouts (e.g., best-of-3), self-consistency helps reduce variance and improve reliability. However, as the number of rollouts increases, low-quality generations accumulate and

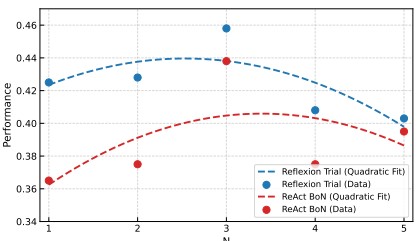

**Figure 8: Impact of inference-time computation on agent performance.** We compare ReAct with a Best-of-$N$ strategy (`DeepSeek-V3.1`) and Reflexion with iterative retries (`o4-mini`).

begin to outweigh the correct ones, while inference time grows almost linearly with the rollout count. This creates a clear trade-off: limited rollouts can stabilize performance and boost accuracy, but excessive repetition ultimately degrades both efficiency and reliability, making adaptive allocation strategies essential for practical deployment.

## 4    RELATED WORKS

**LLM Agents for Scientific Discovery.** Recent advances in LLM agents have demonstrated capabilities such as advanced reasoning Liu et al. (2025b), code-based tool use Ren et al. (2025), and iterative strategies like reflection and planning Liu et al. (2025a). Building on these abilities, early prototypes such as the AI Scientist Lu et al. (2024) and Agent Laboratory Schmidgall et al. (2025) explored end-to-end research automation, from hypothesis generation and experimental design to code execution and report writing. However, these systems were evaluated primarily within machine learning subfields and offered limited validation against real scientific studies. In particular, current work rarely tests agents on realistic multimodal workflows. `MoSciBench` addresses this gap by introducing a principled, domain-diverse benchmark for systematically evaluating LLM agents in multimodal, data-driven scientific discovery.

**Benchmarks for Data-Driven Scientific Discovery.** A growing body of work has introduced benchmarks to evaluate LLM agents in scientific and data-driven workflows Lai et al. (2026); Zhou et al. (2026). Early efforts focused on statistical analysis and AutoML benchmarks Chan et al. (2024), or on code generation from structured tasks Liu et al. (2024). More recently, DiscoveryBench Majumder et al. (2024) and ScienceAgentBench Chen et al. (2024b) have formalized the workflow of data-driven discovery, providing structured tasks, annotated programs, and graded evaluation criteria. Science-Board Sun et al. (2025) further extends this line by creating a realistic multi-domain environment where agents interact with professional software to complete end-to-end scientific workflows. While these contributions mark important progress, most tasks remain either unimodal (e.g., tabular data) or centered on system-level interactions. In contrast, realistic scientific discovery requires integrative reasoning over diverse modalities such as imagery, time series, metadata, and text. *Our `MoSciBench` addresses this gap by introducing the first benchmark explicitly designed for multimodal, data-driven scientific discovery with hypothesis verification grounded in peer-reviewed studies.*

## 5 CONCLUSION

In this work, we introduced `MoSciBench`, the first benchmark for multimodal data-driven scientific discovery, constructed from peer-reviewed studies across five scientific domains. `MoSciBench` formalizes tasks as end-to-end workflows that require cross-modal alignment, scientific computation, and reasoning to verify hypotheses. Through systematic evaluation, we show that current LLM agents struggle with multimodal synthesis, highlighting fundamental gaps in their ability to integrate heterogeneous evidence. By providing scientifically grounded tasks and reproducible evaluations, `MoSciBench` establishes a new challenge space for advancing LLM agents toward more reliable and impactful roles in scientific discovery.

### ACKNOWLEDGMENTS

This work was supported by the National Natural Science Foundation of China (Grant No. 62572417, No.92370204), National Key R&D Program of China (Grant No.2023YFF0725004), the Guangzhou Basic and Applied Basic Research Program under Grant No. 2024A04J3279, Education Bureau of Guangzhou Municipality, and CCF-DiDi GAIA Collaborative Research Funds.

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

# A APPENDIX

## A.1 `MoSciBench` DATA

**Data Source.** Our dataset is curated from peer-reviewed scientific papers across six domains: climate science, biomedical engineering, cheminformatics, health psychology, population genomics, and earth science. Specifically, we replicate workflows and hypotheses from prior works in each domain, including climate science Kitamoto et al. (2024), biomedical engineering Anicai & Shakir (2025), cheminformatics Bushuiev et al. (2024), health psychology Hosseini et al. (2022), population genomics Calafell & Biagini (2019), and earth science Chen et al. (2024a). All datasets and associated assets are released under CC or other permissive open licenses, ensuring accessibility and compliance with data-sharing standards.

**Composition of Instructions.** To illustrate how each task is represented in our benchmark, we provide a detailed explanation of the task instruction schema. Each field defines a specific aspect of the scientific problem, from context and hypothesis to workflow, expected answer format, and evaluation criteria, as shown below:

---

**Task Instruction Schema**

`id` – Unique identifier of the task instance.

`background` – Contextual description of the scientific motivation for the task.

`hypothesis` – The scientific hypothesis to be tested.

`workflow` – Step-by-step instructions outlining the expected analysis procedure.

`gold_hypothesis` – Ground-truth scientific conclusion derived from the data.

`scientific_domain` – The disciplinary area where the task belongs (e.g., biomedical engineering).

`problem_type` – The reasoning category of the task (e.g., descriptive, predictive, causal).

`task_type` – Specific methodological type (e.g., statistical tests).

`domain_knowledge` – Key domain knowledge or definitions needed to perform the task.

`modality` – The data modality used (e.g., time series, tabular, image).

`answer_format` – The expected format of the answer, ensuring consistency across outputs.

`evaluation` – The reference numerical value for evaluation (e.g., percentage of outliers).

`judge_type` – The evaluation criterion (e.g., exact match "=", range-based).

---

## A.2 EXPERIMENTS

### A.2.1 EXPERIMENTAL SETUP

**Agent Frameworks.** At present, there are no dedicated multimodal discovery agents. Existing multimodal foundation models, such as vision–language models or pretrained fusion architectures, are not suitable baselines, as they primarily target images and text. In contrast, `MoSciBench` covers a far richer set of modalities, including long multivariate time series, tabular data, satellite imagery, molecular structures, mass spectra, genotype matrices, and HDF simulation outputs. These formats exceed the input constraints of current models, making direct application infeasible. Consequently, end-to-end LLM agents with tool use remain the only practical approach for executing repository-level discovery workflows in this setting. To ensure fairness and coverage, we benchmark four representative frameworks: (1) **NoDataGuess.** A naive baseline that provides only task descriptions and relies entirely on the LLM's internal memory and reasoning. (2) **ReAct** Yao et al. (2023): Alternates between reasoning and code execution in an iterative loop to refine hypotheses. (3) **DataVoyager.** A modular pipeline with planner, code generator, analysis, and critic components to orchestrate discovery. (4) **Reflexion (Oracle).** Extends CodeGen with oracle feedback and up to three iterative retries for self-improvement. (5) **SelfDebug**. A program-execution agent that iteratively

**Table 6: Performance comparison of Qwen-based LLM agents across domains.** We report accuracy for six scientific domains. "Overall Avg" refers to the macro average across domains.

| Method | Climate Sci. | Biomedical Eng. | Cheminformatics | Health Psych. | Pop. Genomics | Earth Sci. | Overall Avg |
|---|---|---|---|---|---|---|---|
| Qwen3-235B | | | | | | | |
| NoDataGuess | 0.000 | 0.000 | 0.000 | 0.000 | 0.000 | 0.000 | **0.000** |
| ReAct | 0.500 | 0.529 | 0.133 | 0.267 | 0.385 | 0.357 | **0.361** |
| DataVoyager | 0.500 | 0.588 | 0.200 | 0.200 | 0.231 | 0.357 | **0.346** |
| Reflexion | 0.286 | 0.471 | 0.133 | 0.200 | 0.231 | 0.143 | **0.244** |
| Qwen3-Coder | | | | | | | |
| NoDataGuess | 0.000 | 0.000 | 0.067 | 0.067 | 0.000 | 0.000 | **0.022** |
| ReAct | 0.500 | 0.529 | 0.200 | 0.333 | 0.462 | 0.429 | **0.408** |
| DataVoyager | 0.429 | 0.471 | 0.300 | 0.267 | 0.192 | 0.214 | **0.312** |
| Reflexion | 0.500 | 0.588 | 0.200 | 0.133 | 0.154 | 0.143 | **0.286** |

debugs itself by inspecting execution traces, detecting failure symptoms, and rewriting code based on internal error hypotheses. (6) **RAG-ReAct.** A retrieval-augmented variant of ReAct that supplements the agent's reasoning trajectory with external domain knowledge.

### A.2.2 FURTHER EXPERIMENTS ON OTHER BASE MODEL

We further conduct experiments to evaluate the performance of Qwen-based LLM agents across six scientific domains. As shown in Table 6, the ReAct framework consistently outperforms all other methods for both Qwen3–235B and Qwen3–Coder. For Qwen3–Coder, ReAct achieves the highest overall average of 0.408, demonstrating strong effectiveness in climate science (0.500) and biomedical engineering (0.529), while also maintaining competitive results in earth science (0.357). Similarly, for Qwen3–235B, ReAct again leads with an overall average of 0.361, showing notable robustness in population genomics (0.385) and earth science (0.357). These results highlight ReAct as the most reliable framework across domains, underscoring its ability to generalize effectively in complex multimodal scientific settings.

### A.2.3 ERROR ANALYSIS AND AGENT ENHANCEMENT

**Error Analysis.** To characterize agent failures in multimodal scientific discovery, we classify errors into three categories: alignment (conceptual or implementation misalignments), modeling (representation, planning, or computation errors), and reasoning (logical or statistical inference errors). Alignment errors include concept misalignments (e.g., selecting the wrong variable, lead–lag mismatches, or entity mismapping) and implementation misalignments (e.g., faulty joins, missing keys, or unit conversion failures). Modeling errors arise from flawed representations or plans that fail to capture essential scientific relationships, as well as from training or implementation issues, including coding mistakes, unstable optimization, or feature misordering. Reasoning errors encompass logical or statistical inference mistakes (e.g., conflating correlation with causation or misusing significance thresholds) and reporting or output parsing errors (e.g., incorrect answer formats or failed result extraction). This taxonomy clarifies why agents may generate outputs that appear plausible yet deviate from correct multimodal reasoning, and collectively spans the entire agent workflow. We adopt the LLM-as-judge framework to classify each failure instance into these categories, ensuring systematic and reproducible evaluation. Our analysis centers on the best-performing agent, ReAct with the base model o4-mini, as summarized in Figure 4. Alignment errors dominate. The majority of errors are alignment-related (31.8%), such as information mismatches and data processing failures. The root cause lies in the difficulty of integrating datasets across domains and transforming heterogeneous forms, distributions, scales, and resolutions into unified, computable representations while retaining domain-specific information. In comparison, modeling errors account for 15.9% and reasoning errors represent only 3.4%.

### A.2.4 EXTENDED EVALUATION WITH ADDITIONAL AGENTS AND METRICS

To broaden the experimental coverage, we additionally include two baselines that stress code reliability and external knowledge usage: Self-Debug, a code-execution agent with iterative repair capabilities, and RAG-ReAct, a retrieval-augmented variant of ReAct tailored for scientific discovery. Table 3 reports their performance across six scientific domains. Overall, ReAct-style agents continue to excel in producing executable analyses, Self-Debug improves robustness through stepwise correction, and RAG-ReAct shows mixed gains, suggesting that naïvely incorporating external knowledge does not consistently benefit data-driven scientific inference. Table 3 reveals several noteworthy patterns across the six scientific domains. First, although most agents achieve nearly perfect Code Execution Success Rates, their Accuracy (Acc) can be substantially lower. For instance, under `gpt-5-mini`, RAG-ReAct reaches Exec = 1.000 yet its accuracy in Cheminformatics is only 0.067, indicating that the generated code is syntactically valid but scientifically misaligned. Similar trends appear under `Qwen3-30B-A3B`, where DataVoyager attains Exec = 0.875 in Cheminformatics but produces only 0.067 correct answers, showing that executable workflows can still follow incorrect modeling assumptions. Second, Self-Debug illustrates the limits of execution-focused agents. It consistently fixes runtime errors (e.g., Exec = 1.000 in Climate Science under `o4-mini`) yet achieves only moderate accuracy (0.333), because robustness to code failures does not guarantee that the analytical pipeline is conceptually sound. Its strength lies in error recovery rather than hypothesis refinement. Third, ReAct-style agents demonstrate the most balanced performance. Across models, ReAct achieves both high execution reliability (often Exec = 1.000) and competitive accuracy—for example, 0.571 in Climate Science and 0.647 in Biomedical Engineering under `o4-mini`. This suggests that interleaving reasoning with incremental code generation helps maintain scientific coherence and reduces modeling drift. Finally, retrieval-augmented RAG-ReAct exhibits mixed benefits. Although retrieval improves Exec uniformly, its accuracy can drop sharply when external knowledge conflicts with the dataset. For example, under `gpt-5-mini`, RAG-ReAct achieves Exec = 1.000 but accuracy falls to 0.000–0.067 in multiple domains, revealing that naïve retrieval may introduce noise rather than useful inductive priors.

