# OpenReview forum: "Towards Multimodal Data-Driven Scientific Discovery Powered by LLM Agents"
_ICLR.cc/2026/Conference — ICLR 2026 Poster_

### Official Review · Reviewer_BRwp · 2025-10-28

**Soundness:** 3
**Presentation:** 2
**Contribution:** 3
**Rating:** 4
**Confidence:** 4

**Summary:**

This article designs a multi-model benchmark for data-driven scientific discovery, evaluating agents across five categories of discovery questions and seven data modalities. It also tests on state-of-the-art models, analyzes their performance, and proposes the “ReAct + Workflow” method to enhance agent effectiveness.

**Strengths:**

1.  The core contribution of this paper is the introduction of MoSciBench, which is the first benchmark focused on evaluating LLM Agents in performing multimodal data-driven scientific discovery tasks.
2.  MoSciBench itself covers five categories of discovery questions and seven data modalities, demonstrating good comprehensiveness.

**Weaknesses:**

1. The article only shows that the performance of ReAct + Domain Knowledge deteriorates, but does not analyze why the performance worsens. Moreover, what would be the effect if Domain Knowledge and Workflow were added simultaneously?
2. Based on reference [1] and the examples shown in the article, MoSciBench appears to be testing the model's ability to integrate various data for reasoning. However, according to Table 3, the performance of NoDataGuess is very poor; even with o4-mini, it hardly gets any answers correct. It would be beneficial to add an analysis regarding NoDataGuess.
3. Figure 4 is the same as Figure 9. The text does not introduce Figure 4 and directly uses Figure 9 in line 318. The existence of Figure 4 seems to be meaningless.

[1] Who Gets Cited Most? Benchmarking Long-Context Language Models on Scientific Articles

**Questions:**

Table 6 only shows the results for React, but lines 658 - 660 mention that “the ReAct framework consistently outperforms all other methods for both Qwen3–235B and Qwen3–Coder,” which cannot be concluded from Table 6.

---

> ### Author Response · Authors · 2025-11-20
> **Response by Authors [Part 1/2]**
>
> Thank you for your detailed and insightful feedback! We appreciate that the reviewer acknowledged the comprehensiveness of our benchmark design and the value of our agent-performance analyses. Below, we provide a point-by-point response to your concerns.
>
> > W1: "The article only shows that the performance of ReAct + Domain Knowledge deteriorates, but does not analyze why the performance worsens. Moreover, what would be the effect if Domain Knowledge and Workflow were added simultaneously?"
>
> Thank you for raising this important question. **Our findings clearly show two phenomena: (1) adding Domain Knowledge to ReAct leads to accuracy degradation, and (2) combining Domain Knowledge with Workflow Scaffolding further reduces performance**. We summarize the underlying reasons below.
>
> Regarding domain knowledge, although domain knowledge provides high-level scientific heuristics, it does not improve the agent’s ability to ground its reasoning in the actual multimodal data. As shown in Fig. 5, naïve injection of such knowledge introduces (i) contextual noise, which distracts the model from data parsing and cross-modal alignment, and (ii) conflicting priors, which can override the true patterns in the dataset. For instance, in Climate Science tasks, generic priors about typical cyclone–intensity relationships,  can mislead the model when the correct answer depends on case-specific temporal variations.
>
>
> Regarding the simultaneous use of Domain Knowledge and Workflow, we conducted an additional experiment combining both components. As shown in the table below, the overall accuracy drops to 0.437, which is lower than using Workflow alone (0.541) and even lower than the original ReAct baseline (0.484).
>
> | **Workflow Type**                       | **Climate Sci.** | **Biomedical Eng.** | **Cheminformatics** | **Health Psych.** | **Pop. Genomics** | **Earth Sci.** | **Overall** |
> | --------------------------------------- | ---------------- | ------------------- | ------------------- | ----------------- | ----------------- | -------------- | ----------- |
> | **ReAct**                               | 0.571            | 0.647               | 0.333               | 0.533             | 0.462             | 0.357          | **0.484**   |
> | **ReAct + Domain Knowledge**            | 0.500            | 0.588               | 0.267               | 0.667             | 0.385             | 0.286          | **0.449**   |
> | **ReAct + Workflow**                    | 0.714            | 0.706               | 0.333               | 0.533             | 0.462             | 0.500          | **0.541**   |
> | **ReAct + Domain Knowledge + Workflow** | 0.357            | 0.706               | 0.200               | 0.467             | 0.462             | 0.429          | **0.437**   |
>
> Domain Knowledge and Workflow Scaffolding play distinct roles: the former provides **conceptual priors**, while the latter provides **procedural, data-grounded guidance**. When simply concatenated, the verbose domain-knowledge block competes with the workflow instructions, creating instruction-level interference. The model becomes uncertain about which cues to prioritize, pays less attention to the provided multimodal evidence, and tends to overfit high-level priors rather than follow the data-driven workflow. This explains why simultaneous inclusion yields additional performance deterioration.
>
>
> > W2: "Based on reference [1] and the examples shown in the article, MoSciBench appears to be testing the model's ability to integrate various data for reasoning. However, according to Table 3, the performance of NoDataGuess is very poor; even with o4-mini, it hardly gets any answers correct. It would be beneficial to add an analysis regarding NoDataGuess. ([1] Who Gets Cited Most? Benchmarking Long-Context Language Models on Scientific Articles)"
>
>
> Thank you for the helpful suggestion. We agree that analyzing NoDataGuess clarifies what MoSciBench is designed to measure. **NoDataGuess receives only the task description, without any of the provided scientific data, and must rely entirely on the model’s internal priors**. Its extremely low accuracy, even with a strong model such as o4-mini, is therefore expected and highlights a central property of MoSciBench: **the tasks cannot be solved through memorized knowledge or general heuristics; they require grounded reasoning over the actual multimodal evidence.**
>
> Concretely, NoDataGuess often generates answers anchored in broad discipline-specific priors rather than the true data. For example, in Cheminformatics tasks that hinge on specific mass-spectrometry peak relationships, NoDataGuess defaults to generic chemical heuristics and consistently produces incorrect conclusions. This pattern shows that language, only agents systematically fail because MoSciBench problems depend on interpreting numerical patterns, cross-modal correlations, and case-specific data characteristics, not on recalling textbook knowledge.

---

> ### Author Response · Authors · 2025-11-20
> **Response by Authors [Part 2/2]**
>
> > W3: "Figure 4 is the same as Figure 9. The text does not introduce Figure 4 and directly uses Figure 9 in line 318. The existence of Figure 4 seems to be meaningless."
>
>
> We thank the reviewer for noticing the redundancy between Figure 4 and Figure 9.
> The duplication occurred because, due to the page-limit constraint, we originally included Figure 4 to provide a clearer, standalone visualization of the same analysis for easier reference. We appreciate the reviewer’s suggestion and will merge the two figures and remove the redundancy in the camera-ready version to improve clarity and presentation consistency.
>
>
> > Q1: "Table 6 only shows the results for React, but lines 658 - 660 mention that “the ReAct framework consistently outperforms all other methods for both Qwen3–235B and Qwen3–Coder,” which cannot be concluded from Table 6."
>
> Thanks for your careful observations. We have updated Table 6 (Appendix Lines 648–658) to include results for all four agent frameworks under both Qwen3–235B and Qwen3–Coder. As shown in the updated tables below, the results now clearly confirm that ReAct consistently achieves the highest overall performance across both model variants, in line with the statement in the manuscript (Appendix A.2.2). This validates our claim that among the evaluated agent paradigms, ReAct offers the most robust performance in data-driven scientific discovery tasks.
>
> #### Qwen3-235B
> | **Method**       | **Climate Sci.** | **Biomedical Eng.** | **Cheminformatics** | **Health Psych.** | **Pop. Genomics** | **Earth Sci.** | **Overall Avg** |
> |------------------|------------------|----------------------|----------------------|-------------------|-------------------|----------------|-----------------|
> | **NoDataGuess**  | 0.000            | 0.000                | 0.000                | 0.000             | 0.000             | 0.000          | **0.000**       |
> | **ReAct**        | 0.500            | 0.529                | 0.133                | 0.267             | 0.385            | 0.357          | **0.361**       |
> | **DataVoyager**  | 0.500            | 0.588                | 0.200                | 0.200             | 0.231             | 0.357          | **0.346**       |
> | **Reflexion**    | 0.286            | 0.471                | 0.133                | 0.200             | 0.231             | 0.143          | **0.244**       |
>
> #### Qwen3-Coder
> | **Method**       | **Climate Sci.** | **Biomedical Eng.** | **Cheminformatics** | **Health Psych.** | **Pop. Genomics** | **Earth Sci.** | **Overall Avg** |
> |------------------|------------------|----------------------|----------------------|-------------------|-------------------|----------------|-----------------|
> | **NoDataGuess**  | 0.000            | 0.000                | 0.067                | 0.067             | 0.000             | 0.000          | **0.022**       |
> | **ReAct**        | 0.500            | 0.529                | 0.200               | 0.333            | 0.462            | 0.429         | **0.408**       |
> | **DataVoyager**  | 0.429            | 0.471                | 0.300                | 0.267             | 0.192             | 0.214          | **0.312**       |
> | **Reflexion**    | 0.500            | 0.588                | 0.200                | 0.133             | 0.154             | 0.143          | **0.286**       |

---

> > ### Comment · Reviewer_BRwp · 2025-11-23
> > **Response to Authors**
> >
> > I would like to thank the authors for their responses.
> >
> > As previously commented, "MoSciBench appears to be testing the model's ability to integrate various data for reasoning". Current task formulation aligns more closely with information retrieval and integration, which seems more like a multimodal QA task in the scientific field and may not fit the theme of "scientific discovery."
> >
> > Furthermore, I am concerned about the integrity of the initial submission regarding Section A.2.2. The results supporting these conclusions were only provided in the newly added Table 6 during the rebuttal. This indicates that the initial submission asserted conclusions without the necessary experimental evidence. Such a practice suggests the manuscript was premature and incomplete at the time of submission, which undermines the reliability of the reported findings."

---

> > > ### Author Response · Authors · 2025-11-23
> > > **Thanks for Reviewer and Response**
> > >
> > > Thank you for the reviewer’s detailed response. We would like to kindly **clarify a potential misunderstanding** regarding the setting of data-driven scientific discovery. In short, MoSciBench is fundamentally different from multimodal QA because it evaluates an end-to-end, **code-generating and code-executing hypothesis-testing workflow**.
> > >
> > >
> > >
> > > Specifically, given a **scientific hypothesis** and **its associated multimodal dataset**, the agent must generate executable analysis code, run it on raw data, interpret the resulting numerical outputs, and determine whether the hypothesis is supported or refuted. The conclusion is never contained in the prompt; it must be **derived through code-driven validation on the dataset**, mirroring how hypotheses are tested in real scientific practice.
> > >
> > > To illustrate this more concretely, we provide a representative example from MoSciBench:
> > >
> > > **Example (Pattern Discovery Task)**
> > >
> > > * **Hypothesis:** *Global heatwave-affected areas exhibit an upward trend from 2021 to 2023.*
> > >
> > > * **Dataset:** A multimodal climate repository containing global daily temperature-anomaly fields for 2021–2023.
> > >
> > > * **Required Agent Workflow:**
> > >
> > >   1. **Generate analysis code** to compute annual heatwave-affected area from daily temperature anomaly maps.
> > >   2. **Execute the code** on raw data to obtain yearly area estimates.
> > >   3. **Interpret the numerical results** (e.g., 2.1M → 2.9M → 3.4M km²).
> > >   4. **Output a data-supported conclusion** about the trend direction.
> > >
> > > * **Expected Output:** `Trend = upward`
> > >
> > > This example demonstrates that solving a MoSciBench task requires producing and executing a scientific workflow, not retrieving facts or performing multimodal QA, thereby directly evaluating the core ability of LLM agents to perform data-driven scientific discovery.
> > >
> > >
> > >
> > > Regarding the further experiments, the additional Qwen experiments in Table 6 were included only to broaden model coverage and make the analysis more comprehensive; they do not affect or change any conclusions in the original submission.

---

> > > > ### Comment · Reviewer_BRwp · 2025-11-26
> > > >
> > > > Thanks for your clarification. I have raised my score.

---

> > > > > ### Author Response · Authors · 2025-11-27
> > > > > **Thanks for Raising the Score**
> > > > >
> > > > > Dear Reviewer BRwp,
> > > > >
> > > > > Thank you sincerely for your thoughtful feedback and for raising the score. We truly appreciate your recognition. We value your suggestions and will incorporate them in our next revision. Thank you again for your time and constructive comments.
> > > > >
> > > > > Sincerely,
> > > > >
> > > > > Authors

---

### Official Review · Reviewer_mCwP · 2025-11-01

**Soundness:** 3
**Presentation:** 3
**Contribution:** 3
**Rating:** 6
**Confidence:** 3

**Summary:**

This paper introduces MoSciBench, the first benchmark specifically designed for multimodal data-driven scientific discovery powered by LLM Agents. MoSciBench consists of 88 tasks, evaluated through a principled four-stage pipeline, which assesses agents on repository-level tasks that require alignment, modeling, and reasoning across seven data modalities and six scientific domains.

**Strengths:**

1. MoSciBench is the first multimodal benchmark, covering 7 data modalities and 6 scientific domains.

2. MoSciBench identifies the bottleneck of cross-modal alignment of  LLM agents in real-world multimodal scientific tasks. The author presents that over 30% of failures stem from misaligned data rather than flawed reasoning, which is a valuable insight.

3. MoSciBench reveals the importance of data grounding in LLM-based scientific discovery. NoDataGuess approach performing close to 0% accuracy indicates that relying solely on LLMs’ internal knowledge is insufficient for solving scientific problems.

4. By evaluating NoDataGuess, ReAct, Reflexion, and the proposed DataVoyager, the paper reveals critical limitations across current agent architectures.

**Weaknesses:**

1. The baseline coverage is limited. Since all evaluated agents are prompt-based reasoning and code generation frameworks like ReAct and Reflexion, no domain task-specific Agents, multimodal-specific Agents, or retrieval-augmented Agents are included.

2. While alignment errors are identified as the dominant failure mode, the root causes are not explored. Whether this arises from architectural limitations (e.g., lack of explicit alignment modules) or inherent model incapacity is not discussed. Without deeper mechanistic analysis and case studies, the paper does not provide further guidance beyond a generic suggestion of “better alignment is needed”.

3. The evaluation relies exclusively on end-to-end exact match, which may conflate semantically correct solutions with fundamental failures. It is better to report a secondary metric (e.g., partial/subtask credit) or analysis for correct reasoning traces to better disentangle reasoning failure from execution failure.

4. The 1-hour execution cap is not explained. Some failures may be due to time limits rather than methodological flaws.

5. The paper includes several formatting and naming inconsistencies. e.g., in lines 278 and 646, it mixes the spelling of its self-proposing framework “DataVoyager” and “DataVoyage”. In line 295,  there is a missing space before “DeepSeek-V3.1”. In line 689, there is a missing space between the text and the period.

**Questions:**

1. Could you provide a few concrete examples of alignment errors as case studies, and explain what kind of mechanism might cause them to happen?

2. Have you tested whether a longer runtime significantly improves performance?

---

> ### Author Response · Authors · 2025-11-20
> **Response by Authors [Part 1/3]**
>
> Thank you for your detailed and insightful feedback! We appreciate the reviewer’s recognition that MoSciBench is the first multimodal benchmark designed specifically for data-driven scientific discovery, and we are encouraged that our analyses of cross-modal alignment and data grounding were found particularly valuable. Below, we provide a point-by-point response addressing all concerns.
>
> > W1: "The baseline coverage is limited. Since all evaluated agents are prompt-based reasoning and code generation frameworks like ReAct and Reflexion, no domain task-specific Agents, multimodal-specific Agents, or retrieval-augmented Agents are included." & W3: "The evaluation relies exclusively on end-to-end exact match, which may conflate semantically correct solutions with fundamental failures. It is better to report a secondary metric (e.g., partial/subtask credit) or analysis for correct reasoning traces to better disentangle reasoning failure from execution failure."
>
> Thank you for raising this point. We would like to clarify that MoSciBench’s baseline coverage reflects the current state of the literature rather than the benchmark’s limitation. At present, **no domain-specific, multimodal-specific, or retrieval-augmented scientific discovery agents** have been proposed in prior work. Existing agents, including ReAct, Reflexion, and recently introduced single-domain systems, are all built upon prompt-based reasoning and code-generation pipelines, and therefore naturally fall into this paradigm.
> Within this ecosystem, we follow the standard data-driven scientific discovery setting and evaluate four representative agents spanning the major LLM-agent paradigms: (1) NoDataGuess (memory-based reasoning), (2) ReAct (iterative reasoning with tool use),
> (3) DataVoyage (modular planning and analysis), and (4) Reflexion (self-improvement via feedback).
>
> To further broaden coverage, we also evaluate SelfDebug, a more advanced code-execution agent, as well as retrieval-augmented scientific discovery agents RAG-ReAct adapted to the data-driven setting.  For secondary metrics, we report two complementary metrics (Lines 666–672): (1) Code Execution Success Rate (Exec), which measures whether the generated code runs without errors, and (2) Modeling Rationality (MR), scored by LLM-as-judge on a 1–5 scale to assess the scientific validity of the workflow (variable selection, modeling design, statistical soundness).
>
> The results show that agents often demonstrate strong intermediate performance even when their final predictions are incorrect. For example, ReAct with o4-mini achieves only 0.484 accuracy, yet reaches 100% code-execution success and 3.75/5 modeling rationality, indicating that it can produce runnable and scientifically coherent workflows but fails at the final conclusion due to multimodal alignment challenges. DataVoyager exhibits a similar pattern, 0.321 accuracy, but 0.593 execution success and 3.63 rationality, showing that many reasoning steps are correct even though the end-to-end answer is wrong. Overall, these results show that exact-match accuracy remains a strict indicator of final correctness, while the complementary metrics reveal the intermediate reasoning processes. The other additional experiments can be seen in the revised Appendix (Appendix A.2.4).
>
> | **Method**         | **Climate Sci.**     | **Biomedical Eng.**  | **Cheminformatics**  | **Health Psych.**    | **Pop. Genomics**    | **Earth Sci.**       | **Overall**              |
> | ------------------ | -------------------- | -------------------- | -------------------- | -------------------- | -------------------- | -------------------- | ------------------------ |
> |                    | *(Acc / Exec / MR)*  | *(Acc / Exec / MR)*  | *(Acc / Exec / MR)*  | *(Acc / Exec / MR)*  | *(Acc / Exec / MR)*  | *(Acc / Exec / MR)*  | *(Acc / Exec / MR)*      |
> | **NoDataGuess**    | 0.143 / - / -        | 0.000 / - / -        | 0.200 / - / -        | 0.067 / - / -        | 0.077 / - / -        | 0.143 / - / -        | **0.105 / - / -**        |
> | **ReAct**          | 0.571 / 1.000 / 3.50 | 0.647 / 1.000 / 3.76 | 0.333 / 1.000 / 3.93 | 0.533 / 1.000 / 3.80 | 0.462 / 1.000 / 3.92 | 0.357 / 1.000 / 3.57 | **0.484 / 1.000 / 3.75** |
> | **DataVoyager**    | 0.357 / 0.375 / 3.50 | 0.588 / 1.000 / 3.59 | 0.133 / 0.714 / 3.79 | 0.333 / 0.667 / 3.67 | 0.231 / 0.200 / 3.75 | 0.286 / 0.600 / 3.50 | **0.321 / 0.593 / 3.63** |
> | **Reflexion**      | 0.429 / 0.690 / 3.50 | 0.706 / 0.880 / 3.50 | 0.400 / 0.609 / 3.77 | 0.467 / 0.565 / 3.80 | 0.462 / 0.289 / 3.67 | 0.286 / 0.500 / 3.67 | **0.458 / 0.589 / 3.65** |
> | **SelfDebug**      | 0.429 / 0.864 / 3.57 | 0.647 / 0.833 / 3.71 | 0.333 / 0.600 / 3.87 | 0.400 / 0.571 / 3.93 | 0.385 / 0.125 / 3.77 | 0.429 / 0.667 / 3.79 | **0.437 / 0.610 / 3.77** |
> | **RAG-ReAct** | 0.643 / 1.000 / 3.86 | 0.588 / 1.000 / 3.76 | 0.333 / 1.000 / 3.80 | 0.400 / 1.000 / 3.64 | 0.462 / 1.000 / 3.92 | 0.214 / 1.000 / 3.93 | **0.440 / 1.000 / 3.82** |

---

> ### Author Response · Authors · 2025-11-20
> **Response by Authors [Part 2/3]**
>
> > W2: While alignment errors are identified as the dominant failure mode, the root causes are not explored. Whether this arises from architectural limitations (e.g., lack of explicit alignment modules) or inherent model incapacity is not discussed. Without deeper mechanistic analysis and case studies, the paper does not provide further guidance beyond a generic suggestion of “better alignment is needed”." &  Q1: "Could you provide a few concrete examples of alignment errors as case studies, and explain what kind of mechanism might cause them to happen?"
>
>
>
> We thank the reviewer for the insightful comments. We agree that alignment failures likely arise from both intrinsic base-model limitations and the absence of explicit cross-modal alignment mechanisms, and we have conducted targeted experiments in Sections 3.2 and 3.3 to examine both factors.
>
> Regarding base-model capacity (Sec. 3.2): Our model-scaling analysis reveals a clear monotonic trend: stronger LLMs produce stronger agents. o4-mini achieves the highest performance (48.9% with ReAct; 46.6% with Reflexion), DeepSeek-V3.1 performs moderately (36.5% with ReAct), while gpt-5-mini lags significantly (17.4%). This consistent pattern shows that many alignment failures originate from insufficient base-model capability, particularly weaknesses in multimodal description parsing and long-context integration, both essential for grounding hypotheses in heterogeneous scientific data.
>
> Regarding alignment mechanisms (Sec. 3.3, Lines 356–377): To isolate architectural effects, we explicitly inject lightweight human workflow scaffolding, step-level instructions for aligning and interpreting multimodal inputs, into the agent’s context. This intervention produces a substantial improvement in overall accuracy (48.4% → 54.1%), with especially large gains in domains heavily reliant on cross-modal fusion such as climate science (57.1% → 71.4%) and earth science (35.7% → 50.0%). Most importantly, the proportion of alignment errors drops markedly (31.8% → 27.3%), while fully correct solutions increase correspondingly (53.4% → >60%). These results demonstrate that making alignment cues explicit leads to more stable and accurate reasoning, confirming that the absence of explicit alignment modules is indeed a key architectural bottleneck.
>
>
> Here we provide two examples as case studies.
>
> Case Study 1: Predicting Which Physiological Signal Best Predicts Stress (EDA / HR / Temp)
>
> Observed alignment failures:
> 1. Agents ignored the provided survey-derived stress labels and instead invented alternative proxies (e.g., BVP-based scores), indicating a failure to bind the task objective to the correct dataset column.
>
> 2. The task requires mapping stress labels to specific time-windowed sensor segments, but agents treated the modalities as already aligned, failing to infer the link:  labels → time windows → sensor streams.
>
> Mechanistic cause:
> Agents lack mechanisms for multi-file or multi-table schema grounding. Without explicit alignment cues, they rely on memorized heuristics (e.g., “BVP is often used for stress”), causing objective drift and temporal misalignment.
>
>
> Case Study 2: Identifying the French Population with the Highest Heterozygosity
>
> Observed alignment failures:
> 1. Genotype rows in `.bed` must align one-to-one with individual IDs in `.fam` file, since the `.fam` FID provides the correct population label for each individual. However, the agents instead grouped individuals using columns from `france_group_info.csv`, which is only an aggregated overview and not aligned with the per-individual order. This indicates a failure to infer cross-file structural alignment rules such as:
>     `.bed` row i ↔ `.fam` row i ↔ correct population ID,
>     leading the model to incorrectly treat heterogeneous metadata files as interchangeable.
>
> 2. Hallucinated external tools. Some agents proposed workflows invoking PLINK, even though the benchmark requires Python-only computation. This reflects reliance on pretraining templates instead of grounded reasoning about the available data and environment.
>
> Mechanistic cause: Agents do not maintain structured representations linking `.bed/.fam/.bim` formats and thus fail to enforce row-level individual alignment, yielding incorrect grouping and invalid workflows.

---

> ### Author Response · Authors · 2025-11-20
> **Response by Authors [Part 3/3]**
>
> > W4: "The 1-hour execution cap is not explained. Some failures may be due to time limits rather than methodological flaws." & Q2 " Have you tested whether a longer runtime significantly improves performance?"
>
>
>
> Thank you for the questions. Regarding the 1-hour execution cap, we verified that all benchmarked runs finished well within this limit, and no task was terminated due to timeout. Therefore, the cap does not affect correctness; it simply provides a practical and empirically validated balance between computational efficiency and stable execution for large-scale agent benchmarking.
> Regarding whether extending the runtime improves performance, we conducted an empirical analysis of inference-time computation. Specifically, we increased rollout steps and retry counts via two common strategies, Best-of-N sampling for ReAct and iterative retries for Reflexion (results shown in Figure 8). Performance improves slightly up to N = 3, but quickly plateaus or even declines for larger N, exhibiting clear diminishing returns. This indicates that allowing substantially longer runtimes would not meaningfully increase accuracy, and may in fact introduce additional variance and error compounding across extra rollouts.
> Overall, both analyses suggest that the 1-hour cap is sufficient and does not limit agent performance, and extending runtime would not change the conclusions of the benchmark.
>
>
> > W5: "The paper includes several formatting and naming inconsistencies. e.g., in lines 278 and 646, it mixes the spelling of its self-proposing framework “DataVoyager” and “DataVoyage”. In line 295, there is a missing space before “DeepSeek-V3.1”. In line 689, there is a missing space between the text and the period."
>
> We thank the reviewer for carefully pointing out the minor formatting and naming inconsistencies. We have thoroughly checked the manuscript and corrected all such issues, including the inconsistent spelling of “DataVoyager,” missing spaces before “DeepSeek-V3.1,” and punctuation spacing errors. These revisions will be reflected in the revised version.

---

### Official Review · Reviewer_Yaab · 2025-11-01

**Soundness:** 3
**Presentation:** 3
**Contribution:** 3
**Rating:** 4
**Confidence:** 2

**Summary:**

This paper introduces MoSciBench, a benchmark for multimodal, data-driven scientific discovery powered by LLM agents. The benchmark includes six scientific domains, seven data modalities, and five discovery task types, totaling 88 tasks. The authors systematically evaluate several LLM-based agent frameworks and provide a detailed analysis of error sources and limitations. Overall, the paper addresses an important and underexplored problem — assessing AI agents in real-world, multimodal scientific workflows.

**Strengths:**

This paper introduces MoSciBench, a benchmark for multimodal, data-driven scientific discovery powered by LLM agents. The benchmark includes six scientific domains, seven data modalities, and five discovery task types, totaling 88 tasks. The authors systematically evaluate several LLM-based agent frameworks and provide a detailed analysis of error sources and limitations. Overall, the paper addresses an important and underexplored problem — assessing AI agents in real-world, multimodal scientific workflows.

**Weaknesses:**

1. The paper tackles a meaningful and challenging goal — end-to-end scientific discovery from heterogeneous data sources — which is timely and relevant to the emerging intersection of AI agents and scientific reasoning.

2. MoSciBench is well designed, covering a diverse set of domains and modalities. The data curation pipeline is clearly described and appears reproducible.

3. The authors perform an insightful breakdown of error categories (alignment, modeling, reasoning), providing useful diagnostic information for the community.

**Questions:**

1. Although the benchmark aims to emulate “scientific discovery,” most tasks are still formulated as structured, answerable queries with gold labels. The open-ended and hypothesis-generating aspects of real discovery are largely absent.

2. Only a small set of existing agent frameworks are evaluated. It is unclear whether the conclusions generalize beyond the tested models.

3. Reported accuracy numbers (≈50%) are low and not deeply analyzed beyond descriptive statistics. There is limited discussion of why certain domains are more difficult.

4. For evaluation metrics, the reliance on exact-match accuracy is restrictive and might underestimate partial success or reasoning quality.

---

> ### Author Response · Authors · 2025-11-20
> **Response by Authors [Part 1/3]**
>
> Thank you for your detailed and insightful feedback. We appreciate Reviewer’s emphasis on the importance of evaluating end-to-end scientific workflows and their recognition of MoSciBench’s broad coverage across domains, modalities, and hypothesis types. Below, we provide a point-by-point response to each of the reviewer’s concerns.
>
> > Q1: "Although the benchmark aims to emulate “scientific discovery,” most tasks are still formulated as structured, answerable queries with gold labels. The open-ended and hypothesis-generating aspects of real discovery are largely absent."
>
> We appreciate the reviewer’s thoughtful observation. We would like to clarify that MoSciBench is intentionally designed as a **controlled scientific discovery verification benchmark**, rather than a fully open-ended hypothesis-generation benchmark. While tasks are expressed as structured queries with gold labels, this structure is essential for rigorous, reproducible, and automatable evaluation: each task provides real multimodal scientific data and a hypothesis extracted from peer-reviewed studies, and the agent must independently perform data alignment, multimodal fusion, code generation, computation, and conclusion verification.
>
> In contrast, open-ended scientific discovery requires agents to freely propose hypotheses, an evaluation setting that inherently lacks objective ground truth and thus cannot be assessed reliably or at scale. Such tasks are important, but they represent a fundamentally different goal. Our intention with MoSciBench is to establish a **measurable and trustworthy foundation for scientific-reasoning agents** through controlled hypothesis verification, which can later support more open-ended discovery benchmarks.
>
> Importantly, MoSciBench already goes substantially beyond existing benchmarks, which typically involve unimodal data or slice-level reasoning. Each MoSciBench task is formulated as a **cross-modal hypothesis-verification workflow**, requiring the agent to align, integrate, and interpret heterogeneous scientific datasets before performing analytical modeling and reasoning. This design captures a central bottleneck of real-world scientific discovery, **multimodal data alignment and workflow execution**, while maintaining the reproducibility necessary for trustworthy evaluation.

---

> ### Author Response · Authors · 2025-11-20
> **Response by Authors [Part 2/3]**
>
> > Q2: "Only a small set of existing agent frameworks are evaluated. It is unclear whether the conclusions generalize beyond the tested models."
>
> Thank you for raising this point. We would like to clarify that the current literature does not contain any domain-specific, multimodal, or retrieval-augmented agents designed for scientific discovery. Existing frameworks are predominantly text-centric or single-modality and cannot be applied to multimodal scientific workflows without substantial adaptation. To broaden the evaluation beyond the four representative systems in our main experiments, we additionally adapted and tested **SelfDebug** (an advanced program-executing agent) and **RAG-ReAct** (a retrieval-augmented variant) following prior work [1,2] (Lines 661–665). Together, these agents span a wide range of paradigms, iterative reasoning, modular tool use, program synthesis and execution, and retrieval-augmented decision making.
>
> Across all these diverse architectures, we consistently observe the same dominant failure mode: **multimodal alignment errors**, rather than modeling or reasoning errors. This strongly suggests that the primary bottleneck arises from the intrinsic complexity of MoSciBench tasks, aligning heterogeneous scientific modalities, rather than from the specific agent framework or prompting strategy. Moreover, RAG-ReAct exhibits behavior highly consistent with our domain-knowledge-injection experiments. Retrieving external knowledge often introduces contextual noise or conflicting priors, which distract the agent from interpreting multimodal inputs and can contradict the actual patterns in the dataset. As a result, retrieval tends to degrade performance rather than mitigate alignment difficulty, indicating that retrieval alone does not address the core scientific reasoning challenge.
> The full empirical results are provided below.
>
> | **Method** | **Climate Sci.** | **Biomedical Eng.** | **Cheminformatics** | **Health Psych.** | **Pop. Genomics** | **Earth Sci.** | **Overall** |
> | --------------- | ---------------- | ------------------- | ------------------- | ----------------- | ----------------- | -------------- | ----------- |
> | **NoDataGuess** | 0.143 | 0.000 | 0.200 | 0.067 | 0.077 | 0.143 | **0.105** |
> | **ReAct** | 0.571 | 0.647 | 0.333 | 0.533 | 0.462 | 0.357 | **0.484** |
> | **DataVoyager** | 0.357 | 0.588 | 0.133 | 0.333 | 0.231 | 0.286 | **0.321** |
> | **Reflexion** | 0.429 | 0.706 | 0.400 | 0.467 | 0.462 | 0.286 | **0.458** |
> | **SelfDebug** | 0.429 | 0.647 | 0.333 | 0.400 | 0.385 | 0.429 | **0.437** |
> | **RAG-ReAct** | 0.643 | 0.588 | 0.333 | 0.400 | 0.462 | 0.214 | **0.440** |
>
> [1] ScienceAgentBench: Toward Rigorous Assessment of Language Agents for Data-Driven Scientific Discovery
>
> [2] Discoverybench: Towards data-driven discovery with large language models
>
> > Q3: "Reported accuracy numbers (≈50%) are low and not deeply analyzed beyond descriptive statistics. There is limited discussion of why certain domains are more difficult."
>
>
> Thank you for the thoughtful observation. The ≈50% accuracy levels in MoSciBench indeed reflect the intrinsic difficulty of the benchmark. Each task requires the agent to align, fuse, and analyze **multiple heterogeneous scientific modalities**, often combining high-dimensional images, long multivariate time series, structured metadata, and domain-specific measurement formats. This cross-modal integration step is far more demanding than unimodal reasoning and constitutes the central challenge of multimodal scientific discovery.
>
> Our error analysis (Lines 309–323, Figure 4) provides a clear explanation for the observed performance. **Alignment errors are by far the dominant failure mode**, accounting for 31.8% of all errors, compared with only 15.9% for modeling and 3.4% for reasoning. This pattern indicates that the primary bottleneck is not inadequate statistical modeling or incorrect inference, but rather the difficulty of transforming heterogeneous scientific inputs into coherent, computable representations that enable downstream reasoning.
>
> Domain-level differences further support this interpretation. Domains such as Biomedical Engineering and Climate Science achieve higher accuracy because their modalities are relatively structured and standardized (e.g., physiological sensor streams, uniform meteorological formats). In contrast, Cheminformatics, Earth Science, and Population Genomics involve highly heterogeneous, high-dimensional data, mass spectra, genotype matrices, long multivariate time series, and 512×512 HDF images, that require complex, domain-specific alignment. Consistently, these domains show the highest alignment-error rates (e.g., 42.9% in Climate and Earth Science, 38.5% in Population Genomics), confirming that performance variation arises from intrinsic multimodal alignment difficulty rather than simple differences in data length or format.

---

> ### Author Response · Authors · 2025-11-20
> **Response by Authors [Part 3/3]**
>
> > Q4: "For evaluation metrics, the reliance on exact-match accuracy is restrictive and might underestimate partial success or reasoning quality."
>
> Thank you for this insightful suggestion. We fully agree that exact-match accuracy alone cannot capture partial progress or intermediate reasoning quality. However, in multimodal, data-driven scientific discovery tasks, each hypothesis has an **objectively verifiable ground truth**, obtained by executing the underlying scientific computation (e.g., statistical testing, physical simulation, mass-spectrometry processing). For this reason, **exact match remains the only fully rigorous and reproducible criterion** for determining whether an agent has successfully completed the entire end-to-end workflow, data alignment, modeling, code generation, execution, and conclusion validation.
>
> To address the reviewer’s concern, we additionally report two complementary metrics that measure partial success and reasoning quality:
>
> - Code Execution Success Rate (Exec), indicating whether the generated code runs without errors.
> - Modeling Rationality (MR), an LLM-as-judge score (1–5) assessing the scientific validity of the workflow, including variable selection, modeling steps, and statistical reasoning.
>
> These metrics reveal substantial intermediate progress even when exact-match accuracy is limited. For example, **ReAct (o4-mini) attains only 0.484 accuracy but achieves 100% execution success and a strong 3.75/5 rationality score**, showing that the agent constructs runnable and scientifically reasonable workflows but fails at the final step due to misaligned multimodal inputs. DataVoyager exhibits a similar pattern, with lower accuracy (0.321) but meaningful execution success (0.593) and rationality (3.63). These patterns demonstrate that agents often complete much of the workflow correctly and that failures frequently stem from multimodal alignment issues rather than from erroneous modeling or computation.
> Thus, while exact-match accuracy is required for rigorous end-to-end evaluation, our complementary metrics provide a nuanced view of partial success and error sources, offering a deeper understanding of agent capabilities.
>
>
>
> | **Method** | **Climate Sci.** | **Biomedical Eng.** | **Cheminformatics** | **Health Psych.** | **Pop. Genomics** | **Earth Sci.** | **Overall** |
> |---|---|---|---|---|---|---|---|
> |  | *(Acc / Exec / MR)* | *(Acc / Exec / MR)* | *(Acc / Exec / MR)* | *(Acc / Exec / MR)* | *(Acc / Exec / MR)* | *(Acc / Exec / MR)* | *(Acc / Exec / MR)* |
> | **NoDataGuess** | 0.143 / - / - | 0.000 / - / - | 0.200 / - / - | 0.067 / - / - | 0.077 / - / - | 0.143 / - / - | **0.105 / - / -** |
> | **ReAct** | 0.571 / 1.000 / 3.50 | 0.647 / 1.000 / 3.76 | 0.333 / 1.000 / 3.93 | 0.533 / 1.000 / 3.80 | 0.462 / 1.000 / 3.92 | 0.357 / 1.000 / 3.57 | **0.484 / 1.000 / 3.75** |
> | **DataVoyager** | 0.357 / 0.375 / 3.50 | 0.588 / 1.000 / 3.59 | 0.133 / 0.714 / 3.79 | 0.333 / 0.667 / 3.67 | 0.231 / 0.200 / 3.75 | 0.286 / 0.600 / 3.50 | **0.321 / 0.593 / 3.63** |
> | **Reflexion** | 0.429 / 0.690 / 3.50 | 0.706 / 0.880 / 3.50 | 0.400/ 0.609 / 3.77 | 0.467 / 0.565 / 3.80 | 0.462 / 0.289 / 3.67 | 0.286 / 0.500 / 3.67 | **0.458 / 0.589 / 3.65** |

---

> > ### Author Response · Authors · 2025-11-26
> > **Thanks to the Reviewers and Looking Forward to Your Response**
> >
> > Dear Reviewer Yaab,
> >
> > Thank you for your valuable time and insightful comments. We have provided detailed responses and additional results that we believe address your concerns. Specifically, we have included new metrics such as **code execution success rate** and **model rationality**, and added stronger baselines, including **SelfDebug** (an advanced program-executing agent) and **RAG-ReAct** (a retrieval-augmented variant).
> >
> > We would be grateful to hear whether our revisions have sufficiently resolved your concerns. Please let us know if any part of our work remains unclear.
> >
> > Best regards,
> >
> > Authors

---

### Official Review · Reviewer_AMy8 · 2025-11-01

**Soundness:** 3
**Presentation:** 3
**Contribution:** 3
**Rating:** 8
**Confidence:** 4

**Summary:**

This paper introduces MoSciBench, a multimodal benchmark designed for scientific discovery that enables agents to access complete repositories, integrate heterogeneous data, generate and execute code, and reason over results to verify scientific hypotheses. The experiments across 88 tasks reveal that cross-modal alignment is a significant bottleneck, while lightweight workflow scaffolding consistently enhances performance.

**Strengths:**

1.	Unlike previous unimodal benchmarks, MoSciBench explicitly targets multimodal, repository-level discovery, significantly increasing task complexity and realism. This benchmark will be valuable for evaluating the progress of AI agents within the community.
2.	The experiments conducted provide valuable insights into enhancing agents in scientific domains, highlighting areas for further development.

**Weaknesses:**

1.	The ground-truth hypotheses and answers in MoSciBench are derived from peer-reviewed publications. How rigorous is this benchmark? Additionally, if an agent can access search engines, how would that impact its ability to find answers?
2.	For Figure 4, are there significant differences in error distributions among tasks with varying requirements? Which specific data models are particularly prone to failure, and is this related to the length or format of the data?
3.	Did the introduction of lightweight human workflow scaffolding change the proportion of model invocation tools/code execution used? Without this scaffolding, would models demonstrate a need for additional information or behave differently in their outputs?

**Questions:**

identical to the 'weaknesses'

---

> ### Author Response · Authors · 2025-11-20
> **Response by Authors [Part 1/2]**
>
> Thank you for your detailed and insightful feedback! We thank the reviewer for appreciating that MoSciBench moves beyond unimodal or slice-level benchmarks toward realistic multimodal, repository-level scientific discovery and valued our analysis of alignment errors and workflow scaffolding. Below is our point-by-point response to your concerns.
>
> > W1.1: "The ground-truth hypotheses and answers in MoSciBench are derived from peer-reviewed publications. How rigorous is this benchmark?"
>
> MoSciBench is designed with strict scientific rigor: all hypotheses and answers are **directly grounded in peer-reviewed scientific publications**, ensuring full traceability to validated findings rather than heuristic assumptions. Each task is constructed through a standardized four-stage pipeline: data extraction, multimodal preprocessing, instruction formulation, and human verification.
>
> To guarantee annotation quality, we **reproduce the original scientific workflow using an end-to-end executable script.** Tasks whose reproduced results do not match the published conclusions are re-audited or removed. After this filtering, **all released tasks are fully consistent with their peer-reviewed ground truths**, ensuring the benchmark’s validity, transparency, and reproducibility.
>
>
> > W1.2: "Additionally, if an agent can access search engines, how would that impact its ability to find answers?"
>
> Thanks for the insightful question. Allowing search engine access would **not fundamentally reduce the difficulty of MoSciBench.** If an agent retrieves **final scientific conclusions or factual answers**, this would bypass the benchmark’s purpose, as the tasks require deriving hypotheses from the provided multimodal evidence. If it retrieves **modeling workflows or analysis templates**, these may offer structural hints but do not replace the core challenge of aligning, integrating, and interpreting heterogeneous scientific data.
>
>
> To further examine this hypothesis, we explicitly simulated search-like behavior by injecting external domain knowledge or workflow hints into the agent’s context (Lines 356–377). The results show that directly supplying domain knowledge often introduces semantic noise or misalignment and can even degrade performance, while lightweight workflow scaffolding yields small yet consistent gains. These findings demonstrate that external retrieval, whether knowledge-or workflow-oriented, does not overcome the intrinsic multimodal alignment and scientific-reasoning challenges that MoSciBench is designed to assess.

---

> ### Author Response · Authors · 2025-11-20
> **Response by Authors [Part 2/2]**
>
> > W2: "For Figure 4, are there significant differences in error distributions among tasks with varying requirements? Which specific data models are particularly prone to failure, and is this related to the length or format of the data?"
>
> Thanks for the thoughtful question. Across all tasks in MoSciBench, we observe a highly consistent pattern: **alignment errors are the dominant failure mode**, regardless of the task’s analytical requirement. As shown below, alignment errors exceed modeling and reasoning errors in descriptive (32.0%), correlational (44.4%), predictive (30.0%), and pattern-discovery tasks (35.7%), and are tied with reasoning errors in causal-inference tasks (9.1%). This indicates that models primarily struggle to align heterogeneous scientific inputs with the required analytical workflow, rather than with the task type itself.
>
>
> | Problem Type      | Alignment Error (%) | Modeling Error (%) | Reasoning Error (%) |
> | ----------------- | ------------------- | ------------------ | ------------------- |
> | Descriptive       | 32.0                | 12.0               | 4.0                 |
> | Correlational     | 44.4                | 11.1               | 11.1                |
> | Causal Inference  | 9.1                 | 0.0                | 9.1                 |
> | Predictive        | 30.0                | 20.0               | 0.0                 |
> | Pattern Discovery | 35.7                | 21.4               | 7.1                 |
>
>
>
> A domain-level analysis reveals the same trend. Alignment errors remain the predominant failure mode in domains characterized by **large, high-dimensional, and structurally heterogeneous modalities**, such as Climate Science (42.9%), Earth Science (42.9%), and Population Genomics (38.5%). These domains include long multivariate time series, 512×512 HDF matrices, mass spectra, or genotype matrices, modalities that require substantial multimodal alignment rather than posing simple length- or format-driven challenges. In contrast, domains with cleaner, more homogeneous sensor time series or tabular data (e.g., Biomedical Engineering) exhibit fewer alignment issues but proportionally more modeling errors, reflecting a shift from alignment difficulty to domain-specific modeling complexity.
>
>
> | Domain            | Success Rate (%) | Failure Rate (%) | Alignment Error (%) | Modeling Error (%) | Reasoning Error (%) |
> |-------------------|------------------|------------------|----------------------|---------------------|----------------------|
> | Climate Sci.      | 57.1             | 42.9             | 42.9                 | 0.0                 | 0.0                  |
> | Biomedical Eng.   | 64.7             | 35.3             | 23.5                 | 11.8                | 0.0                  |
> | Cheminformatics   | 33.3             | 66.7             | 20.0                 | 26.7                | 20.0                 |
> | Health Psych.     | 53.3             | 46.7             | 26.7                 | 13.3                | 6.7                  |
> | Pop. Genomics     | 46.2             | 53.8             | 38.5                 | 7.7                 | 7.7                  |
> | Earth Sci.        | 35.7             | 64.3             | 42.9                 | 21.4                | 0.0                  |
>
>
> > W3:" Did the introduction of lightweight human workflow scaffolding change the proportion of model invocation tools/code execution used? Without this scaffolding, would models demonstrate a need for additional information or behave differently in their outputs?"
>
> Thank you for the question. In short, introducing lightweight human workflow scaffolding **does not materially change the proportion of tool invocations or code-execution events**. Its main effect is to reduce alignment failures by helping the agent interpret how the multimodal inputs relate to the required analytical workflow. As shown in Figure 6, scaffolding lowers alignment errors from 31.8% → 27.3%, while modeling-related errors and tool-usage frequencies remain essentially unchanged. This indicates that scaffolding improves workflow understanding, not tool dependency.
>
> Without scaffolding, agents more often misinterpret the task objective or invoke tools in ways inconsistent with the intended analysis, such as applying irrelevant preprocessing, selecting inappropriate variables, or choosing statistical procedures that do not correspond to the hypothesis. The workflow scaffolding mitigates these issues by providing high-level procedural cues that clarify modality alignment and the structure of the inference chain. As a result, the reasoning becomes more consistent and scientifically faithful, even though the overall quantity and type of tool calls remain almost the same.

---

> ### Author Response · Authors · 2025-11-25
> **Thanks to the Reviewers and Looking Forward to Your Response**
>
> Dear reviewer AMy8:
>
> We thank you for the precious review time and valuable comments. We have provided corresponding responses and results, which we believe have covered your concerns. We hope to further discuss with you whether or not your concerns have been addressed. Please let us know if you still have any unclear parts of our work.
>
> Best,
>
> Authors

---

### Meta-Review · Area_Chair_jUBF · 2026-01-04

**Summary:**

The reviewers broadly agree that MoSciBench is a timely, well-motivated, and technically solid benchmark targeting an important gap: multimodal, repository-level, data-driven scientific discovery with LLM-based agents. The benchmark is consistently recognized as the first of its kind to require end-to-end workflows involving data alignment, code generation, execution, and hypothesis verification across heterogeneous scientific modalities.

Concerns mainly center on:
* The scope and framing of “scientific discovery” versus hypothesis verification.
* Baseline coverage and generality across agent architectures.
* Evaluation methodology, particularly reliance on exact-match accuracy.
* The depth of mechanistic analysis of alignment failures.
* Some presentation and experimental completeness issues, largely addressed during rebuttal.

The rebuttal substantially strengthened the paper, resolving most objections and leading at least one initially negative reviewer to raise their score. However, the remained concerns of Reviewer Yaab has not been well resolved. I highly suggest the author significantly revising the work to reflect the feedbacks from the reviewers.
I think this work represents a meaningful and timely contribution that will be valuable for the community studying LLM agents in scientific workflows, there is space for significant improvement for the future work.
I am recommending an acceptance but I wouldn't mind if the paper gets rejected.

**Reviewer Concerns:**

Some reviewers argue the benchmark focuses more on hypothesis verification than open-ended discovery, limiting coverage of hypothesis generation. (Reviewers Yaab).

The addressed reviewers' concerns are:
* limited baseline diversity (raised by Reviewers Yaab and mCwP) which has partially solved as the newly added baselines SelfDebug and RAG-ReAct.
* restricted evaluation metrics (Reviewers Yaab, mCwP) like exact-match accuracy, and the authors have proposed two new metrics Execution Success Rate and the Modeling Rationality.
* Limited mechanistic depth in initial alignment analysis (Reviewers mCwP) and the authors provided more analysis in depth during rebuttal.

**Reviewer Scores:**

Beside the changed scores, I think the reviewers would changed their scores to 4, 6, 6, 8, where Reviewer Yaab would not have raised their score since the authors have not fully addressed their concerns. The reason is two-fold. On one hand, the authors discussed the scope of the proposed benchmark on the scientific discovery, yet they have not consider the full complexity of this scenario. Reviewers have agreed that benchmarking this challenging task is not easy, while their concerns about its simplified setting of this benchmark has not been well addressed. On the other hand, the concern about the restricted evaluation metrics have been partially resolved, the authors have not yet well explained the results of the new evaluation. While there were legitimate concerns regarding scope, baselines, and evaluation rigor, the rebuttal substantially strengthened the paper.

---

### Decision · Program_Chairs · 2026-01-26

Accept (Poster)